# Sparse Parameterization for Epitomic Dataset Distillation

**Xing Wei[1]    Anjia Cao[1]    Funing Yang[1]    Zhiheng Ma[2]***
[1]School of Software Engineering, Xi'an Jiaotong University
[2]Shenzhen Institute of Advanced Technology, Chinese Academy of Sciences
weixing@mail.xjtu.edu.cn   zh.ma@siat.ac.cn
{caoanjia7, moolink}@stu.xjtu.edu.cn

## Abstract

The success of deep learning relies heavily on large and diverse datasets, but the storage, preprocessing, and training of such data present significant challenges. To address these challenges, dataset distillation techniques have been proposed to obtain smaller synthetic datasets that capture the essential information of the originals. In this paper, we introduce a Sparse Parameterization for Epitomic datasEt Distillation (SPEED) framework, which leverages the concept of dictionary learning and sparse coding to distill epitomes that represent pivotal information of the dataset. SPEED prioritizes proper parameterization of the synthetic dataset and introduces techniques to capture spatial redundancy within and between synthetic images. We propose Spatial-Agnostic Epitomic Tokens (SAETs) and Sparse Coding Matrices (SCMs) to efficiently represent and select significant features. Additionally, we build a Feature-Recurrent Network (FReeNet) to generate hierarchical features with high compression and storage efficiency. Experimental results demonstrate the superiority of SPEED in handling high-resolution datasets, achieving state-of-the-art performance on multiple benchmarks and downstream applications. Our framework is compatible with a variety of dataset matching approaches, generally enhancing their performance. This work highlights the importance of proper parameterization in epitomic dataset distillation and opens avenues for efficient representation learning. Source code is available at https://github.com/MIV-XJTU/SPEED.

## 1   Introduction

Deep learning has achieved remarkable success across diverse domains, thanks to its ability to extract insightful representations from large and diverse datasets [1–5]. Nevertheless, the storage, preprocessing, and training of these massive datasets introduce significant challenges that strain storage and computational resources. In response to these challenges, dataset distillation techniques have arisen as a means to distill a more compact synthetic dataset that encapsulates the pivotal information of the original dataset. The central concept of dataset distillation is the extraction of an epitomic representation, capturing the core characteristics and patterns of the original dataset while minimizing storage demands. By doing so, deep learning models trained on the distilled dataset can attain performance levels similar to those trained on the original dataset but with significantly reduced storage and training costs. This approach opens new horizons for applications requiring cost-effective storage solutions and expedited training times, such as neural architecture search [6–8] and continual learning [9–11].

The success of dataset distillation heavily relies on two essential factors: proper parameterization of the synthetic dataset and an effective design of the matching objective to align it with the original

---

*Zhiheng Ma is the corresponding author.

37th Conference on Neural Information Processing Systems (NeurIPS 2023).

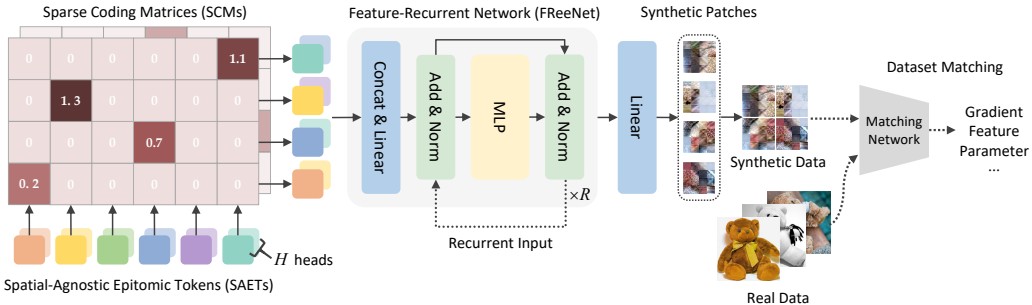

Figure 1: **SPEED Overview.** We take spatial-agnostic epitomic tokens as the shared dictionary of the dataset and perform multi-head sparse combinations to synthesize instance-specific features. Subsequently, we utilize feature-recurrent blocks to generate hierarchical representations for non-linear synthesis of image patches, while reusing the sparse features. In this way, we sparsely parameterize the dataset, alleviating the storage burden and producing highly representative synthetic images.

dataset. While many dataset distillation methods have primarily focused on optimizing the matching objective [12–18], the critical aspect of synthetic dataset parameterization has often been overlooked. Typically, these methods employ a naive image-independent parameterization approach, where each learnable parameter basis (synthetic image) is optimized independently. Although some studies have recognized the inefficiency of image-independent parameterization and explored the mutual coherence and relationship between different synthetic images to improve compression efficiency [19–22], none of the previous methods have fully considered the spatial redundancy that exists within individual images and between different images.

In this paper, we present an efficient parameterization framework, named Sparse Parameterization for Epitomic datasEt Distillation (SPEED), which addresses the aforementioned limitations. SPEED leverages principles from representation learning paradigms, including convolutional neural network (CNN) [23, 1], vision transformer (ViT) [3], dictionary learning [24–30], and sparse coding [31–36].

SPEED introduces Spatial-Agnostic Epitomic Tokens (SAETs) that are shared among all synthetic image patches, and employs the Sparse Coding Matrices (SCMs) to select the most significant tokens. Subsequently, these selected tokens are assembled sequentially to form higher-level representations that facilitate the reconstruction of synthetic patches via non-linear mapping. To further minimize storage requirements, we propose a Feature-Recurrent Network (FReeNet) that utilizes recurrent blocks to generate hierarchical features, with SAETs and SCMs shared by all blocks, while leveraging a multi-head mechanism to enhance feature diversity.

In comparison to previous methods, our approach demonstrates significant advantages in handling high-resolution real-world datasets with substantial spatial redundancy. Notably, it achieves outstanding performance on ImageNet subsets, surpassing most of the previous state-of-the-art methods [21, 37] by achieving an average improvement of 11.2% with 1 image per class storage space. Additionally, our sparse parameterization approach exhibits superior performance on unseen network architectures, outperforming previous state-of-the-art approaches [19, 21, 22], and even surpassing our own full-element baseline, highlighting the potential of sparse representation for storage efficiency and improved generalization abilities.

Our method demonstrates competitive results across three standard dataset distillation benchmarks, such as surpassing the previous state-of-the-art [22] on CIFAR100 by 6.0% and on TinyImageNet by 10.9% when using 1 image per class storage space. It also exhibits strong performance on downstream applications, such as continual learning. Furthermore, our framework is compatible with multiple existing matching objectives [13, 16, 17], generally enhancing their performance through the use of our sparse parameterization strategy.

In summary, our work highlights the importance of proper parameterization in epitomic dataset distillation and introduces the SPEED framework as a solution. We showcase its superiority in handling high-resolution datasets, achieving exceptional performance on benchmarks and downstream

applications. Our framework not only enhances storage efficiency but also improves generalization capabilities, opening new avenues for efficient representation learning in deep learning applications.

## 2    Method

The purpose of dataset distillation [12] is to learn a synthetic dataset $\mathcal{S} = \{(\tilde{X}_i, y_i)\}_{i=1}^N$ that is much smaller in size than the original dataset $\mathcal{T} = \{(X_i, y_i)\}_{i=1}^M$, i.e., $N \ll M$, while minimizing the loss of information. Formally, previous methods optimize the synthetic dataset by minimizing various matching objectives, all of which can be expressed using the following formulation:

$$\mathcal{S}^* = \arg\min_{\mathcal{S}} \mathbb{E}_{\theta \sim \Theta} \Big[ \mathcal{D}\big(\varphi(\mathcal{T}, \theta), \varphi(\mathcal{S}, \theta)\big) \Big], \tag{1}$$

where $\Theta$ represents the distribution used for initializing the network parameters, $\theta$ parameterize the training network, $\mathcal{D}(\cdot, \cdot)$ is the dataset matching metric, $\varphi(\cdot)$ maps the dataset to other informative spaces (e.g. gradient [13], feature [17, 18], and parameter spaces [16]).

However, rather than exclusively focusing on matching objectives, this paper introduces a universal parameterization method for synthetic datasets that can be seamlessly integrated with most existing matching objectives. The naive parameterization method, which optimizes each synthetic image $\tilde{X}$ independently [12–14, 16–18], fails to leverage shared information between images, resulting in unnecessary redundancy. In the following sections, we will present our parameterization framework, which decomposes the synthetic dataset into Spatial-Agnostic Epitomic Tokens, Sparse Coding Matrices, and a Feature-Recurrent Network. This framework significantly mitigates redundancy within and between images, irrespective of the spatial locations of features.

### 2.1    Spatial-Agnostic Recurrent Parameterization

Taking inspiration from the Vision Transformer (ViT) [3], we propose a more efficient parameterization approach applied at the patch level, utilizing cascade non-linear combinations to reduce complex and fine-level redundancy. To further minimize the storage footprint, we share Sparse Coding Matrices (SCMs) and Spatial-Agnostic Epitomic Tokens (SAETs) across all recurrent blocks. The main approach can be formulated as follows:

$$\tilde{X}_i = \Phi_\phi(E, A_i). \tag{2}$$

Here $\tilde{X}_i = [\tilde{x}_i^{(1)}, \tilde{x}_i^{(2)}, ..., \tilde{x}_i^{(J)}] \in \mathbb{R}^{L \times J}$, and $\tilde{x}_i^j \in \mathbb{R}^L$ represents the $j$-th patch of $\tilde{X}_i$, which is flatten into a vector. $L$ equals to the product of the patch height, width, and number of channels, and $J$ is the total patch number. Similar to ViT, we divide the synthetic image into non-overlapping rectangular patches. $E = [e_1, e_2, ..., e_K] \in \mathbb{R}^{D \times K}$ is the Spatial-Agnostic Epitomic Tokens (SAETs) shared by all synthetic patches, where $D$ is the feature dimension, and $K$ is the total number of tokens. $A_i = [a_i^{(1)}, a_i^{(2)}, ..., a_i^{(J)}] \in \mathbb{R}^{K \times J}$ is the Sparse Coding Matrix (SCM) for the $i$-th synthetic image, where $a_i^{(j)} \in \mathbb{R}^K$ is the specific coding vector for patch $\tilde{x}_i^{(j)}$. $A_i$ will be further sparsified and saved in a storage-efficient format. $\Phi_\phi(\cdot)$ is a non-linear recurrent transformer-style network that maps SAETs and the SCM to a synthetic image. Its learnable parameters are denoted as $\phi$. This network is referred to as the Feature-Recurrent Network (FReeNet), which is described in detail below.

**Feature-Recurrent Network.**    In the Feature-Recurrent Network (FReeNet), each recurrent block shares the same SAETs and SCMs for the sake of parameter efficiency. However, this shared approach can lead to a lack of diversity in the resulting representations, as a single pool of SAETs must model multi-scale features. Drawing inspiration from the multi-head mechanism introduced in the transformer architecture, we introduce the concept of "multi-head" SAETs, aiming to strike a balance between storage efficiency and feature diversity. Initially, the original SAETs are split along the feature dimension to create multiple SAET pools, denoted as $\{E^h\}_{h=1}^H$, where $E^h \in \mathbb{R}^{\frac{D}{H} \times K}$. Additionally, each pool is assigned an independent SCM, denoted as $\{A_i^h\}_{h=1}^H$, where $A_i^h \in \mathbb{R}^{K \times J}$. The coding matrix $A_i^h$ will undergo further sparsification to select the most significant tokens. We refer to the mechanism that combines the multi-head SAETs and SCMs as Multi-Head Sparse Coding (MHSC), which can be formulated as follows:

$$\text{MHSC}_r \left( \{E^h\}_{h=1}^H, \{A_i^h\}_{h=1}^H \right) = W_r[E^1 A_i^1, E^2 A_i^2, ..., E^H A_i^H] + b_r, \tag{3}$$

where $W_r \in \mathbb{R}^{D \times D}$ is a linear projection and $b_r$ is the bias, both of which are specific to each recurrent block and not shared across blocks. Using MHSC as the central component, we can construct the FReeNet in a recurrent manner, with SAETs and SCMs shared across different scales:

$$
\begin{aligned}
Z_r' &= \text{LN}_r^1(\text{MHSC}_r\left(\{E^h\}_{h=1}^H, \{A_i^h\}_{h=1}^H\right)) + Z_{r-1}), \quad r = 1, 2, ..., R, \\
Z_r &= \text{LN}_r^2(\text{MLP}_r(Z_r') + Z_r'), \qquad\qquad\qquad\qquad r = 1, 2, ..., R, \\
\tilde{X}_i &= W Z_R + b,
\end{aligned}
\tag{4}
$$

where $R$ is the total number of recurrent blocks, and $Z_R \in \mathbb{R}^{D \times J}$ is the output of the last block. $Z_0$ is initialized with a zero matrix. $W \in \mathbb{R}^{L \times D}$ and $b$ make up the final linear projection layer, and $\tilde{X}_i \in \mathbb{R}^{L \times J}$ is the output synthetic image, which is then rearranged into its original shape. MLP stands for the multi-layer perceptron. In our implementation, we set the MLP to have one hidden layer with the same dimension as the input, and incorporate layer normalization (LN) [38] and residual connection [2] in each block. It is worth noting that the parameters other than the SAETs and SCMs are not shared between different blocks to ensure that each block processes a different scale of features. Despite the use of shared SCMs across different blocks, our analysis demonstrates that SCMs still occupy the majority of the stored parameters. Therefore, we introduce a sparsification method to further enhance the storage efficiency of SCMs, motivated by the theories and techniques of sparse coding [31, 32]. In the subsequent section, we will reuse $\tilde{X}_i = \Phi_\phi(\{E^h\}_{h=1}^H, \{A_i^h\}_{h=1}^H)$ to refer to the multi-head implementation of the FReeNet, without any ambiguity.

## 2.2 Training Objective and Feature Sparsification

Since the $\ell_0$ norm is not differentiable and difficult to optimize [39–42], we instead adopt the $\ell_1$ norm as the sparsity penalty function. By promoting sparsity in solutions, it can effectively remove redundant features [40, 43–45]. Our optimization objective can be expressed as follows:

$$
\mathcal{S} = \left\{ (\Phi_\phi(\{E^h\}_{h=1}^H, \{A_i^h\}_{h=1}^H), y_i) \right\}_{i=1}^N,
$$

$$
\underset{\{E^h\}_{h=1}^H, \{\{A_i^h\}_{h=1}^H\}_{i=1}^N, \phi}{\arg\min} \mathbb{E}_{\theta \sim \Theta} \left[ \mathcal{D}\left( \varphi(\mathcal{T}, \theta), \varphi(\mathcal{S}, \theta) \right) \right] + \lambda \sum_{i=1}^N \sum_{h=1}^H ||A_i^h||_1,
\tag{5}
$$

where $|| \cdot ||_1$ is the $\ell_1$ norm of a matrix, $\lambda$ controls the amount of regularization. Using this approach, we decompose synthetic images into a multi-head SAET $\{E^h\}_{h=1}^H$ and network parameters $\phi$ that are shared by all synthetic images, and a multi-head SCM $\{A_i^h\}_{h=1}^H$ for each synthetic image.

**Feature Sparsification with Global Semantic Preservation.** Sparse codes allow for the ranking of features [33, 46], with higher coefficients indicating more important features. Therefore, we can select the most influential sub-parameters of SCM to achieve the desired sparsity $||A_i^h||_0 \leq k$. This process reduces storage inefficiency while preserving the global semantics of synthetic images. Moreover, all existing matching objectives [13, 16–18] optimize synthetic images using a specific training network, but these synthetic images are then used to train various agnostic network architectures. By pruning unnecessary features of synthetic images, we can further enhance their generalization ability on unseen network architectures. We experimentally validate this claim in Sec. 3.3. Specifically, given a learned SCM $A \in \mathbb{R}^{K \times J}$, we assign a binary mask $B \in \mathbb{R}^{K \times J}$ to it:

$$
B[i,j] = \begin{cases} 1, & A[i,j] \in \text{topk}(\text{abs}(A)) \\ 0, & \text{otherwise} \end{cases}, \quad \bar{A} = B \odot A,
\tag{6}
$$

where $\text{topk}(\cdot)$ obtains the largest $k$ elements, and $\text{abs}(\cdot)$ takes the absolute value of each element of the input matrix. We operate a Hadamard product on the two to preserve the top-$k$ efficient elements of the SCM, *i.e.*, select the most critical epitomic features. By operating on each learned SCM, we receive $\{\{\bar{A}_i^h\}_{h=1}^H\}_{i=1}^N$, which can be directly applied to synthesize images.

For the compressed storage of SCMs, we adopt the widely-used coordinate (COO) format, utilizing the `uint8` data type to store the row and column coordinates of non-zero elements as the size of the sparse matrix is always smaller than or equal to $256 \times 256$ in our implementation. Consequently, the storage needed for each non-zero element is 1.5 times that of a single `float32` tensor. In this way, the storage complexity of SCMs can be greatly reduced from $O(NHKJ)$ to $O(NHk)$, where $k \ll KJ$. The algorithm of our approach is summarized in Alg. 1.

**Algorithm 1** Sparse Parameterization for Epitomic Dataset Distillation (SPEED).

---

**Input:** $\mathcal{T}$: original dataset; $N$: total number of synthetic images; $H$: number of heads; $k$: expected number of non-zero elements of SCM; SPARSIFY$(\cdot, \cdot)$: feature sparsification.

1: Randomly initialize SAETs $\{E^h\}_{h=1}^H$, SCMs $\{\{A_i^h\}_{h=1}^H\}_{i=1}^N$, and the parameters $\phi$ of FReeNet
2: **for each** distillation step... **do**
3:      Get a random initialized training backbone for the matching strategy
4:      Construct the synthetic dataset: $\mathcal{S} = \left\{ (\Phi_\phi(\{E^h\}_{h=1}^H, \{A_i^h\}_{h=1}^H), y_i) \right\}_{i=1}^N$
5:      Inner optimization of the training backbone with respect to the matching strategy (if necessary)
6:      Compute the objective function combined matching loss with sparsity penalty as Eq. (5)
7:      Optimize $\{E^h\}_{h=1}^H, \{\{A_i^h\}_{h=1}^H\}_{i=1}^N, \phi$ with respect to the objective function
8: **end for**
9: **for** $i = 1$ **to** $N$ **do**
10:      **for** $h = 1$ **to** $H$ **do**
11:          Select most significant features: $\bar{A}_i^h \leftarrow$ SPARSIFY$(A_i^h, k)$, according to Eq. (6)
12:          Convert $\bar{A}_i^h$ to compressed storage format
13:      **end for**
14: **end for**

**Output and Save:** $\{E^h\}_{h=1}^H, \{\{\bar{A}_i^h\}_{h=1}^H\}_{i=1}^N$ and $\phi$.

---

**Storage Analysis.** Our storage budget is constrained by the upper bound determined by the storage requirements of the original synthetic images. Specifically, for a given budget of $n$ images per class (IPC) [12, 13] with $c$ total classes, the maximum budget is limited to $cnLJ$, where $LJ$ is the storage requirement for a single synthetic image. Therefore, the following inequality must be satisfied:

$$\underbrace{DK}_{\text{SAETs: } \{E^h\}_{h=1}^H} + \underbrace{1.5NHk}_{\text{SCMs: } \{\{\bar{A}_i^h\}_{h=1}^H\}_{i=1}^N} + \underbrace{R(3D^2 + 7D) + L(D+1)}_{\text{FReeNet: } \phi} \le \underbrace{cnLJ}_{\text{Budget: IPC} = n}. \tag{7}$$

Noticing that SAETs and FReeNet are shared by all synthetic images, and $1.5Hk \ll LJ$, we can synthesize a much larger amount of synthetic images than the original one. For instance, considering the IPC 1 storage budget on CIFAR100, *i.e.*, $c = 100$ and $n = 1$, we set $D = 96$, $K = 64$, $H = 3$, $k = 48$, $R = 2$, $L = 48$, and $J = 64$, resulting in a more informative synthetic dataset with a size of $N = 1100$ (11 final images for each class).

## 3 Experiments

In this section, we first evaluate our method and compare it with previous work. Then, we conduct generalization experiments and perform ablation studies. More results and detailed values for hyper-parameters will be extensively discussed in the appendix. To quantify the performance and guarantee the fairness of the comparison, we use the default `Conv-InstanceNorm-ReLU-AvgPool` ConvNet with 128 channels as our training backbone, consistent with previous methods. We adopt trajectory matching [16] as our default matching objective.

### 3.1 Comparisons

**Standard Benchmarks.** We first conduct experiments on three standard dataset distillation benchmark datasets: CIFAR10 [48] and CIFAR100 [48] at a resolution of $32 \times 32$, and TinyImageNet [49] at a resolution of $64 \times 64$. To adhere to the standard protocol, we employ a 3-layer ConvNet and a 4-layer ConvNet for training and evaluating CIFAR and TinyImageNet, respectively. As illustrated in Tab. 1, SPEED achieves highly competitive results on all three datasets. Remarkably, we achieve a test accuracy of $40.0\%$ and $26.9\%$ on CIFAR100 and TinyImageNet, respectively, using 1 image per class (IPC) storage space, representing improvements of $6.0\%$ and $10.9\%$ over the previous state-of-the-art [22].

**ImageNet Subsets.** We evaluate the performance of SPEED on the high-resolution ImageNet [50] subsets and achieve new state-of-the-art results, as shown in Tab. 2. Consistent with [16, 21], we split

| | Dataset | CIFAR10 | | | CIFAR100 | | | TinyImageNet | | |
|---|---|---|---|---|---|---|---|---|---|---|
| | IPC | 1 | 10 | 50 | 1 | 10 | 50 | 1 | 10 | 50 |
| Coreset | Random | 14.4±2.0 | 26.0±1.2 | 43.4±1.0 | 4.2±0.3 | 14.6±0.5 | 30.0±0.4 | 1.4±0.1 | 5.0±0.2 | 15.0±0.4 |
| | Herding | 21.5±1.3 | 31.6±0.7 | 40.4±0.6 | 8.4±0.3 | 17.3±0.3 | **33.7±0.5** | **2.8±0.2** | 6.3±0.2 | **16.7±0.3** |
| | K-Center | **23.3±0.9** | **36.4±0.6** | **48.7±0.3** | **8.6±0.3** | **20.7±0.2** | 33.6±0.4 | 2.7±0.2 | **7.8±0.4** | 16.7±0.4 |
| | Forgetting | 13.5±1.2 | 23.3±1.0 | 23.3±1.1 | 4.5±0.3 | 9.8±0.2 | - | 1.6±0.1 | 5.1±0.2 | 15.0±0.3 |
| Matching | DC [13] | 28.3±0.5 | 44.9±0.5 | 53.9±0.5 | 12.8±0.3 | 25.2±0.3 | 32.1±0.3 | - | - | - |
| | DSA [14] | 28.8±0.7 | 52.1±0.5 | 60.6±0.5 | 13.9±0.3 | 32.3±0.3 | 42.8±0.4 | - | - | - |
| | KIP [47] | **49.9±0.2** | 62.7±0.3 | 68.6±0.2 | 15.7±0.2 | 28.3±0.1 | - | - | - | - |
| | DM [17] | 26.0±0.8 | 48.9±0.6 | 63.0±0.4 | 11.4±0.3 | 29.7±0.3 | 43.6±0.4 | 3.9±0.2 | 12.9±0.4 | 24.1±0.3 |
| | TM [16] | 46.3±0.8 | 65.3±0.7 | 71.6±0.2 | 24.3±0.3 | 40.1±0.4 | **47.7±0.2** | 8.8±0.3 | 23.2±0.2 | **28.0±0.3** |
| | FRePo [37] | 46.8±0.7 | **65.5±0.4** | **71.7±0.2** | **28.7±0.1** | **42.5±0.2** | 44.3±0.2 | **15.4±0.3** | **25.4±0.2** | - |
| | Parameters / Class | 3,072 | 30,720 | 153,600 | 3,072 | 30,720 | 153,600 | 12,288 | 122,880 | 614,400 |
| Param. | IDC [19] | 50.0±0.4 | 67.5±0.5 | 74.5±0.1 | - | 44.8±0.2 | - | - | - | - |
| | HaBa [21] | 48.3±0.8 | 69.9±0.4 | 74.0±0.2 | 33.4±0.4 | 40.2±0.2 | 47.0±0.2 | - | - | - |
| | RTP [22] | **66.4±0.4** | 71.2±0.4 | 73.6±0.5 | 34.0±0.4 | 42.9±0.7 | - | 16.0±0.7 | - | - |
| | SPEED (Ours) | 63.2±0.1 | **73.5±0.2** | **77.7±0.4** | **40.0±0.4** | **45.9±0.3** | **49.1±0.2** | **26.9±0.3** | **28.8±0.2** | **30.1±0.3** |
| | Whole Dataset | | 84.8±0.1 | | | 56.2±0.3 | | | 37.6±0.4 | |

Table 1: Comparisons with previous dataset distillation and coreset selection methods on standard benchmarks. "Matching" refers to dataset distillation methods with specific matching objectives. "Param." refers to dataset distillation methods with synthetic data parameterization. The bold **numbers** represent the highest accuracy achieved in each category.

| Dataset | ImageNette | | ImageWoof | | ImageFruit | | ImageMeow | | ImageSquawk | | ImageYellow | |
|---|---|---|---|---|---|---|---|---|---|---|---|---|
| IPC | 1 | 10 | 1 | 10 | 1 | 10 | 1 | 10 | 1 | 10 | 1 | 10 |
| TM [16] | 47.7±0.9 | 63.0±1.3 | 28.6±0.8 | 35.8±1.8 | 26.6±0.8 | 40.3±1.3 | 30.7±1.6 | 40.4±2.2 | 39.4±1.5 | 52.3±1.0 | 45.2±0.8 | 60.0±1.5 |
| FRePo [37] | **48.1±0.7** | **66.5±0.8** | **29.7±0.6** | **42.2±0.9** | - | - | - | - | - | - | - | - |
| Parameters / Class | 49,152 | 491,520 | 49,152 | 491,520 | 49,152 | 491,520 | 49,152 | 491,520 | 49,152 | 491,520 | 49,152 | 491,520 |
| HaBa [21] | 51.9±1.7 | 64.7±1.6 | 32.4±0.7 | 38.6±1.3 | 34.7±1.1 | 42.5±1.6 | 36.9±0.9 | 42.9±0.9 | 41.9±1.4 | 56.8±1.0 | 50.4±1.6 | 63.0±1.6 |
| SPEED (Ours) | **66.9±0.7** | **72.9±1.5** | **38.0±0.9** | **44.1±1.4** | **43.4±0.6** | **50.0±0.8** | **43.6±0.7** | **52.0±1.3** | **60.9±1.0** | **71.8±1.3** | **62.6±1.3** | **70.5±1.5** |
| Whole Dataset | 87.4±1.0 | | 67.0±1.3 | | 63.9±2.0 | | 66.7±1.1 | | 87.5±0.3 | | 84.4±0.6 | |

Table 2: Comparisons with previous methods on high-resolution ImageNet subsets.

ImageNet into 6 subsets, namely, ImageNette, ImageWoof, ImageFruit, ImageMeow, ImageSquawk, and ImageYellow, each consisting of 10 classes with resolutions of $128 \times 128$. And a 5-layer ConvNet is employed as the model for both training and evaluation.

Notably, our results achieved with IPC 1 storage space are highly competitive with the previous state-of-the-art results [21, 37] obtained with IPC 10. Specifically, we only need to employ 10% of their parameters to achieve similar or even better performance. Compared to the previous state-of-the-art [21] using the same IPC 1 storage space, our approach exhibits an average improvement of 11.2% across all subsets. Moreover, we maintain a substantial lead for IPC 10 storage space. For instance, we achieve 71.8% accuracy on ImageSquawk, which is a 15.0% improvement over the previous state-of-the-art [21]. These outstanding outcomes are attributed to our design of sharing SAETs among patches, which enables SPEED to be more effective in reducing spatial redundancy as data resolution increases.

**Continual Learning.** Owing to its expressive representations and finer-grained image construction, SPEED has the ability to synthesize an informative dataset. Synthetic images of each class are of exceptional quality, thereby contributing to the dataset's overall richness and relevance. Following the DM [17] setup based on GDumb [51], we conduct continual learning experiments on CIFAR100 with IPC 20 storage space and use the default ConvNet and ResNet18 for evaluation. We randomly divide the 100 classes into 5 learning steps, that is, 20 classes per step. As illustrated in Fig. 2, SPEED maintains the highest test accuracy at all steps for both evaluation networks.

## 3.2 Generalization

**Universality to Matching Objectives.** SPEED exhibits adaptability to multiple existing matching objectives, including those presented in [13, 16, 17], and can be directly integrated with them. In line with [52], we evaluate a diverse set of architectures, including the default ConvNet, ResNet [2], MLP, and ViT [3]. The results and comparisons to corresponding baselines on CIFAR10 are presented in Tab. 3. For instance, when using MLP for evaluation under the IPC 1 budget, SPEED yields a

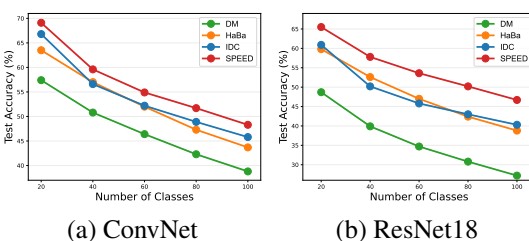
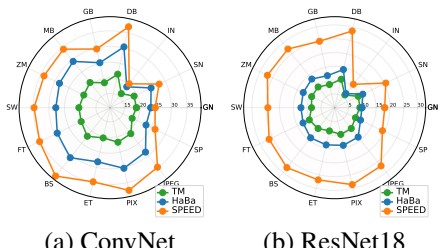

| (a) ConvNet | (b) ResNet18 | (a) ConvNet | (b) ResNet18 |
|---|---|---|---|

Figure 2: Test accuracy of continual learning. SPEED maintains the best performance on all steps.

Figure 3: Robustness of the synthetic dataset on different architectures. Each direction represents each type of corruption.

| | Evaluation | ConvNet | | | MLP | | | ResNet18 | | | ViT | | |
|---|---|---|---|---|---|---|---|---|---|---|---|---|---|
| Matching | IPC | 1 | 10 | 50 | 1 | 10 | 50 | 1 | 10 | 50 | 1 | 10 | 50 |
| Gradient | DC [13] | 29.3±0.4 | 51.0±0.6 | 56.8±0.4 | 29.0±0.5 | 34.1±0.4 | 31.6±0.6 | 27.4±0.7 | 44.0±1.4 | 45.9±1.4 | 28.1±1.1 | 34.4±0.4 | 30.1±0.5 |
| | w. SPEED | 48.5±0.3 | 52.6±0.3 | 59.8±0.4 | 35.5±0.2 | 39.0±0.4 | 41.5±0.3 | 42.5±0.5 | 47.5±0.5 | 55.0±0.3 | 30.4±0.6 | 38.4±0.4 | 39.0±0.3 |
| | Gain | **+19.2** | **+1.6** | **+3.0** | **+6.5** | **+4.9** | **+9.9** | **+15.1** | **+3.5** | **+9.1** | **+2.3** | **+4.0** | **+8.9** |
| Distribution | DM [17] | 26.5±0.4 | 47.6±0.6 | 62.0±0.3 | 10.0±0.6 | 34.4±0.3 | 40.5±0.4 | 20.6±0.5 | 38.2±1.1 | 52.8±0.4 | 20.5±0.5 | 34.4±0.5 | 45.2±0.4 |
| | w. SPEED | 45.0±0.4 | 62.0±0.3 | 66.4±0.3 | 34.8±0.3 | 42.2±0.5 | 46.3±0.4 | 40.7±0.8 | 57.6±0.5 | 65.2±0.3 | 28.4±0.6 | 43.2±0.2 | 48.9±0.3 |
| | Gain | **+18.5** | **+14.4** | **+4.4** | **+24.8** | **+7.8** | **+5.8** | **+20.1** | **+19.4** | **+12.4** | **+7.9** | **+8.8** | **+3.7** |
| Trajectory | TM [16] | 44.2±1.2 | 63.7±0.4 | 70.3±0.6 | 10.4±0.5 | 30.8±0.6 | 38.5±0.3 | 34.2±1.4 | 45.2±1.4 | 60.0±0.7 | 21.5±0.4 | 33.6±0.6 | 47.7±0.6 |
| | w. SPEED | 63.2±0.1 | 73.5±0.2 | 77.7±0.4 | 34.1±0.2 | 44.4±0.4 | 47.7±0.2 | 53.9±0.7 | 69.5±0.4 | 76.4±0.3 | 37.5±0.8 | 51.5±0.3 | 54.5±0.5 |
| | Gain | **+19.0** | **+9.8** | **+7.4** | **+23.7** | **+13.6** | **+9.2** | **+19.7** | **+24.3** | **+16.4** | **+16.0** | **+17.9** | **+6.8** |

Table 3: Universality to different matching objectives. SPEED is compatible with a variety of matching objectives and brings significant accuracy improvements over the corresponding baseline methods. All experiments are conducted with DSA augmentation [14].

significant improvement of 24.8% and 23.7% in accuracy compared to the distribution [17] and trajectory [16] matching baselines, respectively. Our experiments reveal that the improvements in cross-architecture accuracy tend to be more significant than those on the ConvNet. For instance, with an IPC 10 budget for trajectory matching [16], we achieve a 24.3% gain on ResNet18. This outcome further showcases the strengths of our sparse parameterization approach in terms of generalization.

**Cross-Architecture Performance.** The main purpose of dataset distillation is to distill a synthetic dataset that is effective on various even unseen architectures. In this study, we evaluate the cross-architecture performance of our method by comparing it with previous synthetic data parameterization approaches [19, 21, 22] on CIFAR10, using IPC 10 storage space. The results presented in Tab. 4 demonstrate that SPEED continues to

| Method | ConvNet | MLP | ResNet18 | ViT |
|---|---|---|---|---|
| IDC [19] | 67.5±0.5 | 41.4±0.2 | 62.9±0.6 | 47.9±0.8 |
| HaBa [21] | 69.9±0.4 | 35.4±0.4 | 60.2±0.9 | 42.2±0.6 |
| RTP [22] | 71.2±0.4 | 27.2±0.2 | 67.5±0.1 | 35.7±0.4 |
| **SPEED** | **73.5±0.2** | **44.4±0.4** | **69.5±0.4** | **51.5±0.3** |

Table 4: Comparision of cross-architecture generalization evaluation. SPEED exhibits outstanding leads on all architectures.

outperform other methods significantly in terms of generalization across unseen architectures. As an illustration, our method achieves an accuracy of 51.5% on ViT, which represents a 3.6% improvement over the previous state-of-the-art [19]. Although various existing dataset distillation matching objectives tend to overfit the training network, we address this challenge by pruning unnecessary features through sparse parameterization.

**Robustness to Corruption.** To explore the out-of-domain generalization of our synthetic dataset, we conduct experiments on CIFAR100-C [53]. In detail, we evaluate on ConvNet and ResNet18, using the synthetic dataset trained under the IPC 1 budget. Fig. 3 shows the average accuracy of 14 types of corruption under 5 levels respectively. Compared with the previous methods, we achieve better performance under all kinds of corruption. Especially on ResNet18, SPEED outperforms previous methods significantly, achieving almost *double* the test accuracy under every corruption scenario. This demonstrates the generalization and robustness benefits brought by sparse parameterization.

### 3.3 Ablation Study

**Size of $k$ for Feature Sparsification.** Our total storage parameters are positively correlated with the size of $k$, and the representation ability of the SCM is also related to this $k$-winner. Therefore, we

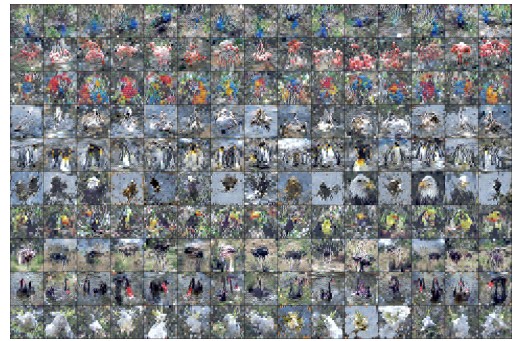 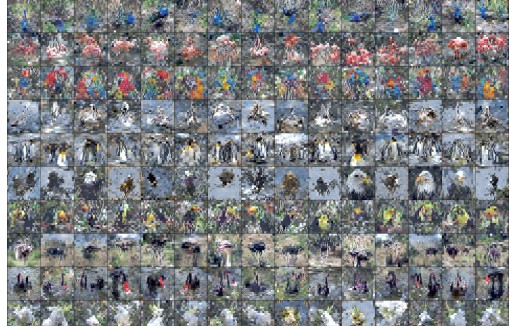

(a) Before feature sparsification        (b) After feature sparsification ($\mathrm{density} = 0.3\%$)

Figure 4: ImageSquawk: synthetic data samples before and after feature sparsification. The process of sparsification does not lead to a significant loss of essential information, resulting in a high degree of similarity between the two images.

| $k$ | # Param | Budget | ConvNet | MLP | ResNet18 | ViT |
|---|---|---|---|---|---|---|
| full | 15M | - | 74.0±0.2 | 44.8±0.3 | 69.0±0.4 | 50.8±0.5 |
| 96 | 575K | - | **74.6±0.3** | **44.9±0.1** | 69.3±0.7 | **51.8±0.5** |
| 48 | 307K | ✓ | 73.5±0.2 | 44.4±0.4 | **69.5±0.4** | 51.5±0.3 |
| 24 | 173K | ✓ | 68.2±0.3 | 35.9±0.7 | 66.4±0.5 | 45.2±0.4 |
| 12 | **106K** | ✓ | 57.8±0.3 | 24.8±0.4 | 59.0±0.4 | 36.7±0.9 |

Table 5: Different $k$ for feature sparsification.

| $R$ | $N/c$ | ConvNet | MLP | ResNet18 | ViT |
|---|---|---|---|---|---|
| 1 | 18 | 20.7±0.4 | 4.5±0.2 | 17.3±0.5 | 9.9±1.1 |
| 2 | 17 | **26.9±0.3** | **6.0±0.3** | **21.3±0.6** | **14.9±0.3** |
| 3 | 15 | 25.4±0.1 | 5.3±0.1 | 19.5±0.6 | 13.9±0.2 |

Table 6: Depth of FReeNet.

| $K$ | $N/c$ | ConvNet | ResNet18 |
|---|---|---|---|
| 32 | 11 | 38.5±0.2 | 29.8±0.5 |
| 64 | 11 | **40.0±0.4** | **29.9±0.3** |
| 96 | 10 | 37.3±0.1 | 29.3±0.4 |
| 128 | 10 | 38.2±0.3 | 29.6±0.4 |

Table 7: Number of SAETs.

| $D$ | $N/c$ | ConvNet | ResNet18 |
|---|---|---|---|
| 48 | 13 | 38.0±0.1 | 29.0±0.4 |
| 72 | 12 | 38.7±0.3 | 29.8±0.5 |
| 96 | 11 | **40.0±0.4** | **29.9±0.3** |
| 144 | 7 | 35.5±0.4 | 26.7±0.3 |

Table 8: Dimension of SAETs.

| $H$ | $N/c$ | ConvNet | ResNet18 |
|---|---|---|---|
| 1 | 33 | 38.7±0.1 | **32.2±0.4** |
| 2 | 16 | 39.3±0.4 | 30.5±0.3 |
| 3 | 11 | **40.0±0.4** | 29.9±0.3 |
| 4 | 8 | 35.9±0.4 | 27.4±0.8 |

Table 9: Head of SAETs.

discuss the preference for the value of $k$. Specifically, for cross-architecture evaluation on CIFAR10 with IPC 10 storage space, we set $k$ to $\{4096, 96, 48, 24, 12\}$, as shown in Tab. 5 (Budget denotes whether the storage budget is met). We observe that taking a moderate value of $k$ can effectively improve the cross-architecture generalization ability of the synthetic dataset while maintaining high test accuracy on the homogeneous network, *i.e.*, the training backbone. When $k$ is set to 96, the overall accuracy exceeds that of the result obtained without feature sparsification, thereby demonstrating the effective removal of inefficient features and the improvement in generalization. In order to meet the storage budget ($10 \times 10 \times 48 \times 64 \approx 307$K), we finally set $k$ to 48, which maintains competitive performance.

**Feature Sparsification Visualizations.** To compare the representation capabilities of SCMs after feature sparsification, we provide some visualization results on ImageSquawk, as shown in Fig. 4. We note that the visualization results using only the sparsified features are highly similar to the corresponding original ones. Differences are noticeable only in a few patches, which are insignificant for classification and mainly consist of meaningless backgrounds. This implies that synthetic images can be constructed effectively using only a few prominent features.

**Trade-off between Quality and Quantity.** Eq. (7) illustrates that SPEED's parameters consist of three parts. The parameters of SAETs and FReeNet are shared among all synthetic images, while the parameters of SCMs are specific to each synthetic image. Thus, under the same budget, increasing shared parameters will lead to a decrease in the number of synthetic images. We conduct ablation experiments on CIFAR100 with IPC 1 storage space to study the trade-off between quality (more shared parameters) and quantity (more synthetic images). Tab. 7-9 study the size of SAETs. As can be seen, keeping a moderate number of synthetic images is important, *i.e.*, $N/c > 10$. A lack of diversity caused by too few images can impact the performance. However, once a certain number of images is reached, further increasing the quantity can lead to a decrease in the quality of each image and slightly reduce performance. In general, our performance is insensitive to the size of SAETs

and outperforms the previous state-of-the-art [22] in multiple settings, except when the number of synthetic images is too small. Tab. 6 studies the depth of FReeNet, we conduct ablation experiments on TinyImageNet with IPC 1 storage space. Although using only one block can maximize the number of synthetic images per class ($N/c$), the lack of hierarchical features severely compromises the model's representational capacity, resulting in poor image quality. However, having too many blocks can result in increased reconstruction overhead. We find that FReeNet with two blocks already achieves a great compromise between performance and computation cost.

**Effects of Increasing Image Resolution.** To study the effectiveness of our method in higher resolution scenarios, we performed experimental investigations on both the distribution matching [17] baseline and our method, using the ImageNette dataset with image sizes of 128×128 and 256×256. In line with the practices of previous methods that handle higher resolution images with deeper networks, we increased the depth of the ConvNet to 6 for the 256×256 image size.

|  | 128×128 | 256×256 | Gain |
|---|---|---|---|
| DM [17] | 28.6±0.6 | 29.5±1.1 | +0.9 |
| w. SPEED | 53.5±1.2 | 57.7±0.9 | +4.2 |
| Gain | **+24.9** | **+28.2** | **+3.3** |

Table 10: Effects of increasing image resolution on ImageNette.

As shown in Tab. 10, when the resolution increases from 128×128 to 256×256, the gain brought by SPEED to the baseline also amplifies from 24.9% to 28.2%. The results demonstrate that our method achieves more substantial improvements when applied to higher-resolution images.

## 4 Related Work

**Dataset Distillation.** Dataset distillation, proposed by Wang *et al.* [12], aims to learn a smaller synthetic dataset so that the test performance of the model on the synthetic dataset is similar to that of the original dataset. For better matching objectives, Zhao *et al.* [13] present a single-step matching framework, encouraging the result gradient of the synthetic dataset and the original dataset to be similar, further extended by [15, 19, 54, 55]. Subsequently, Cazenavette *et al.* introduce TM [16] to alleviate the cumulative error problem of single-step matching, which inspires a series of work [56–58]. To avoid the expensive computational overhead brought by complex second-level optimization, Zhao *et al.* [17] suggest DM, a distribution matching approach, and Wang *et al.* [18] explicitly align the synthetic and real distributions in the feature space of a downstream network. There are also some methods based on kernel ridge regression [59, 47, 60, 37], which can bring out a closed-form solution for the linear model, avoiding extensive inner loop training.

In terms of synthetic data parameterization, Kim *et al.* introduce IDC [19], using downsampling strategies to synthesize more images under the same storage budget. Zhao *et al.* [20] propose to synthesize informative data via GAN [61, 62]. Deng *et al.* [22] have explored how to compress datasets into bases and recall them by linear combinations. Liu *et al.* [21] propose a dataset factorization approach, utilizing image bases and hallucinators for image synthesis. Lee *et al.* [63] further factorize the dataset into latent codes and decoders. These parameterization methods are designed to find shareable or low-resolution image bases. Still, they either only consider the connections between synthetic images [21, 22, 63], or do not reduce the redundancy inside the synthetic image thoroughly [19]. So the storage space of their image bases is still proportional to the dataset resolution, bringing out unsatisfactory performance on high-resolution datasets. SPEED is a universal synthetic data parameterization framework, in which the distillation and construction are spatial-agnostic. It allows for joint modeling correlations between and within the synthetic images, leading to lower storage redundancy and finer-grained image synthesis, thus performing competently on high-resolution datasets.

**Sparse Coding.** Sparse coding calls for constructing efficient representations of data as a combination of a few high-level patterns [31–36]. It has been proven to be an effective approach in the field of image reconstruction and image classification [64–70]. Typically, a dictionary of basis functions (e.g. wavelets [71] or curvelets [72]) is used to decompose the image patches into coefficient vectors. By imposing sparsity constraints on the coefficient vectors, efficient sparse representations of the image patches can be obtained. In addition, the dictionary is not necessarily fixed, it can also be learned to adapt to different tasks [25–28], and the penalty function can also be varied [33, 73, 74]. Prior sparse coding research has primarily focused on compressing individual images [75, 76], with an emphasis on achieving high compression ratios while minimizing perceptual distortion. However, there has been limited exploration of how to apply sparse coding to compress entire datasets in a

way that enhances the training of downstream neural networks. Our work demonstrates that theories and techniques in sparse coding can provide valuable inspiration for developing dataset distillation methods.

**Coreset Selection.** Coreset selection identifies a representative subset of the original dataset [77–79, 10]. Its objective is in line with the goals of dataset distillation and can be utilized to tackle challenges such as continual learning [10, 80, 81] and active learning tasks [82]. This technique typically performs better when the storage budget is relatively large, while dataset distillation demonstrates superior performance under extremely limited storage budgets [52].

## 5   Conclusion and Limitations

In this paper, we introduce SPEED, an efficient and generalizable solution for dataset distillation that offers the following merits: First, the spatial-agnostic epitomic tokens distilled by our method are not only shared between the different classes but also shared among patches of every synthetic image, regardless of their spatial locations. Such efficient modeling enables us to perform well on high-resolution datasets with much less spatial redundancy. Second, the proposed feature-recurrent network promotes hierarchical representation in an efficient recurrent manner, resulting in more informative synthetic data. Finally, the proposed feature sparsification mechanism improves both the storage efficiency and the generalization ability. SPEED achieves outstanding performance on various datasets and architectures through extensive experiments.

**Limitations.** Similar to the previous parameterization methods [19–22], decomposing the original synthetic datasets into different components will slightly increase distilling costs and incur reconstruction overhead. We alleviate this issue by designing the network to be as lightweight as possible. While the dataset synthesized by SPEED offers better privacy protection for users compared to the original dataset, there remains a possibility of privacy leakage.

## Acknowledgements

This work was supported in part by the National Natural Science Foundation of China under Grant 62006183 and Grant 62206271, in part by the National Key Research and Development Project of China under Grant 2020AAA0105600, and in part by the Shenzhen Key Technical Projects under Grant JSGG20220831105801004, CJGJZD20220517141605014, and JCYJ20220818101406014.

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

# I Experiment Details

**Datasets and Preprocessing.** We evaluate our methods on the following datasets: i) **CIFAR10 [48]:** A standard image dataset consists of 60,000 32×32 RGB images in 10 different classes, including airplane, automobile, bird, cat, deer, dog, frog, horse, ship, and truck. For each class, 5000 images are used for training and 1000 images are used for testing. ii) **CIFAR100 [48]:** CIFAR100 contains 100 classes. It has a training set with 50,000 images and a testing set with 10,000 images. iii) **TinyImageNet [49]:** A 64×64 image dataset with 200 classes. Each class has 500 images for training and 50 images for testing. iv) **ImageNet [50] subsets:** High resolution (128×128) datasets from ILSVRC2012 [50]. ImageNette (assorted objects) and ImageWoof (dog breeds) are designed for easy and hard classification tasks [83]. ImageFruit, ImageMeow, ImageSquawk, and ImageYellow [16] consist of fruits, cats, birds, and yellowish things, respectively. All the above subsets contain 10 classes.

For fair comparisons, we follow the previous methods [16, 21], adopting ZCA whitening on CIFAR10 and CIFAR100 with the default Kornia [84] implementation, and no ZCA whitening is used on TinyImageNet and ImageNet subsets.

**Evaluation Settings.** The objective of dataset distillation is to learn a synthetic dataset that can be utilized across a range of network structures. This is achieved by matching the raw and synthetic datasets with the aid of a specific network, with the expectation that can generalize to unseen network structures. In accordance with the evaluation presented in [52], we consider a diverse set of architectures, as shown below:

• **ConvNet [1, 85, 86]:** The standard architecture used for both distilling and evaluating synthetic datasets in previous distillation work. The default network contains three 3×3 convolution layers, each followed by 2×2 average pooling and instance normalization. The hidden embedding size is set to 128. There are around 320K trainable parameters. For TinyImageNet, the number of layers is increased to 4 for improved performance, as suggested in previous work [16, 17]. Similarly, we increase it to 5 for ImageNet subsets, following [16].

• **MLP:** The simple MLP architecture is applied for evaluation, which includes 3 fully connected layers, and the width is set to 128. For CIFAR10, CIFAR100, and TinyImageNet, the MLP has 411K, 423K, and 2M trainable parameters, respectively.

• **ResNet18 [2]:** We also evaluate synthetic datasets on the commonly used ResNet architecture with 4 residual blocks. In each block, there are 2 convolution layers followed by ReLU activation and instance normalization (IN) [87]. The number of trainable parameters is around 11M.

• **ViT [3]:** Vision Transformer applies standard transformer [88] on non-overlapping image patches, demonstrating the variants of transformers can also be a competitive alternative to CNNs. We take it as one of the architectures for evaluating synthetic datasets. There are around 10M trainable parameters in the adopted implementation of ViT.

Following the mainstream evaluation settings, we train 5 randomly initialized evaluation networks on the synthetic dataset, using SGD optimizer, where the momentum and weight decay are set to 0.9 and 0.0005, respectively.

**Hyper-Parameters.** For CIFAR10 (32×32), CIFAR100 (32×32), and TinyImageNet (64×64), we fix the default number of patches $J$ to 64, and increase it to 256 for ImageNet subsets (128×128). Then, the default patch dimension $L$ can be directly derived.

In terms of SAETs, which has been studied in Sec. 3.3, we set the number $K$ according to the resolution of the dataset (e.g. we set $K$ to 256 for ImageNet subsets, and decrease it to 128 and 64 for TinyImageNet and CIFAR100, respectively). When the budget is enlarged, we expand the dimension of SAETs $D$ with its head $H$ increased together, while guaranteeing that their divisor $D/H \leq K$, so that each head $E^h$ is a complete (or overcomplete) dictionary. We always build the FReeNet with shallow blocks, which provides sufficient nonlinear capabilities while making it lightweight enough (e.g. we set 1 block on CIFAR10 with IPC 1 storage space which is excessively stringent, and set 2 blocks for all experiments on TinyImageNet and ImageNet subsets).

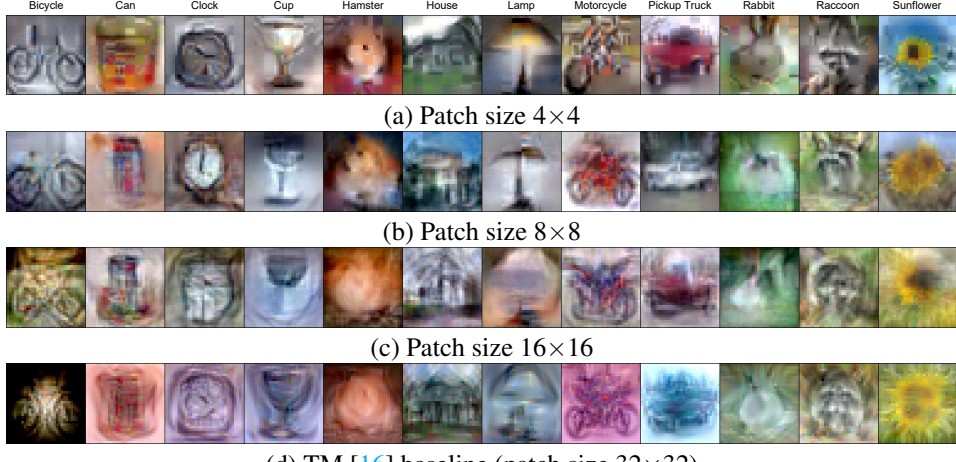

(a) Patch size 4×4

(b) Patch size 8×8

(c) Patch size 16×16

(d) TM [16] baseline (patch size 32×32)

Figure I: Synthetic data samples under different configurations of patch size. A small patch size can capture intricate details such as the black eye patch of the hamster, truck tires, and rabbit ears.

The number of non-zero elements $k$ is always set to a value that ensures the matrix density $k/JK < 5\%$, then the number of images $N$ can be computed according to Eq. (7). The setting of the penalty weight $\lambda$ depends on the number of images, heads, and patches, satisfying $\lambda \approx 0.064/NHJ$.

**Compute Resources.**    Our experiments were run on a mixture of RTX 3090, RTX A6000, and A800 GPUs. The GPU memory consumption is mainly dependent on that of the matching objective and is slightly higher than its baseline, due to the larger data amount. For instance, our experiments based on gradient matching [13] and distribution matching [17] need one 24GB 3090 GPU. In terms of trajectory matching [16], the VRAM usage will be higher, ranging from one 24GB 3090 GPU to six 80GB A800 GPUs.

## II   Additional Results and Analyses

**Patch Size.**    The patch size plays an important role in determining the granularity of our distillation modeling. Let each patch $\tilde{x}$ be of $P \times P$ resolution and contains $C$ channels, then its dimension $L$ will be $P^2C$. Tab. I presents the results on CIFAR100 with IPC 1 storage space. We observe that setting a relatively small patch size can lead to better performance, as it reduces redundancy within synthetic images and enables finer-grained synthesis. Therefore, we set a default patch size of 4×4 for the 32×32 resolution datasets (CIFAR10 and CIFAR100) and increase it appropriately when the resolution of the dataset increases. For instance, we adopt a default patch size of 8×8 for TinyImageNet (64×64).

| Patch size | $N/c$ | ConvNet | MLP | ResNet18 | ViT |
|---|---|---|---|---|---|
| 4 × 4 | 11 | **40.0±0.4** | **15.5±0.2** | **29.9±0.3** | **20.7±0.5** |
| 8 × 8 | 10 | 39.3±0.3 | 14.6±0.3 | 29.4±0.5 | 20.2±0.2 |
| 16 × 16 | 7 | 37.3±0.3 | 12.9±0.3 | 27.1±0.8 | 16.5±0.2 |

Table I: Patch size. Small patch sizes allow finer-grained modeling.

Furthermore, we provide a visualization of synthetic images with varying patch sizes, as depicted in Fig. I. The utilization of small patches can facilitate the expression of finer-grained features, leading to synthetic images that contain more classifiable information. For instance, when a 4×4 patch is employed, it can capture intricate details such as the black eye patch of the hamster, in addition to more noticeable features like truck tires and rabbit ears. Also, sparse parameterization allows for efficient storage of background information with limited significance by allocating fewer non-zero elements in SCMs to represent it.

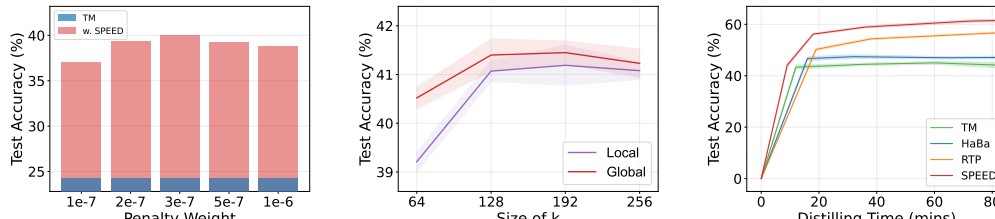

Figure II: **Left:** Effects of different penalty weight $\lambda$ on test accuracy. TM [16] denotes the baseline method of the trajectory matching objective with naive image-independent parameterization. A moderate weight can trade off sparsity and accuracy. **Middle:** Comparison of different feature sparsification strategies. Global sparsification performs better in various settings, especially when the storage budget is stringent. **Right:** Test accuracy and distilling time.

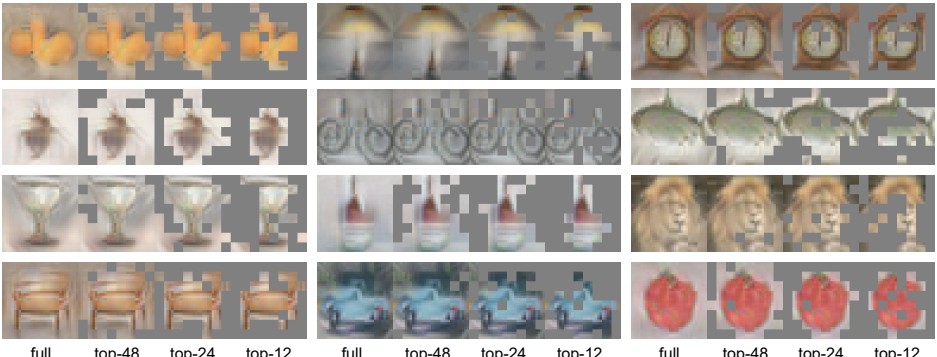

Figure III: Visualization of our SCMs with varying values of $k$ (patch corresponds to zero column vector in its SCMs is marked grey). SPEED learns to efficiently allocate more storage for foreground objects that contain more classifiable information. Therefore, a small $k$ still allows for salient features of objects to be synthesized well.

**Sparsity Penalty Weight** $\lambda$. In SPEED, the sparsity penalty term constrains the overall scale of elements in SCMs. Ablation studies of the penalty weight $\lambda$ are conducted on CIFAR100 with IPC 1 storage space, as shown in Fig. II (left). For these experiments, we fix the number of sparsified features to 48. If the penalty weight is too small, our SCM tends to comprise a higher number of elements with larger absolute values, which necessitates more features to be maintained during feature sparsification, thereby guaranteeing the test performance. It consequently implies a larger storage burden. On the other hand, setting the penalty weight too high may limit the representation capability required for synthesizing informative images. Therefore, we prefer the moderate penalty weight, which allows for sufficient combinations of required epitomic tokens while efficiently eliminating inefficient ones without the risk of losing important information.

**Local Feature Sparsification.** To assess the efficacy of our feature sparsification with global semantic preservation on SCMs, we introduce the contrast, *local feature sparsification*, that attempts to store an equal number of non-zero elements for each patch, *i.e.*, each column of SCM has the same number of non-zero elements. Consequently, local feature sparsification constructs all patches using the same number of parameters (keep $k/J$ elements for each column of SCM), while the feature sparsification with global semantic preservation does not (keep $k$ elements for whole SCM). We evaluate the performance of these two methods using different values of $k$, while maintaining the other hyper-parameters constant.

As illustrated in Fig. II (middle), we conduct tests on CIFAR100 with $k$ set to $\{64, 128, 192, 256\}$, where the SCM size is $64 \times 64$. Accordingly, the local feature sparsification strategy preserves $\{1, 2, 3, 4\}$ features for each synthetic patch (e.g. when $k$ is set at 128, $J = 64$, then each patch is represented by the 2 most prominent SAETs). In all cases, the global feature sparsification outperforms the local one. Notably, the advantages of the global approach are more apparent when the storage

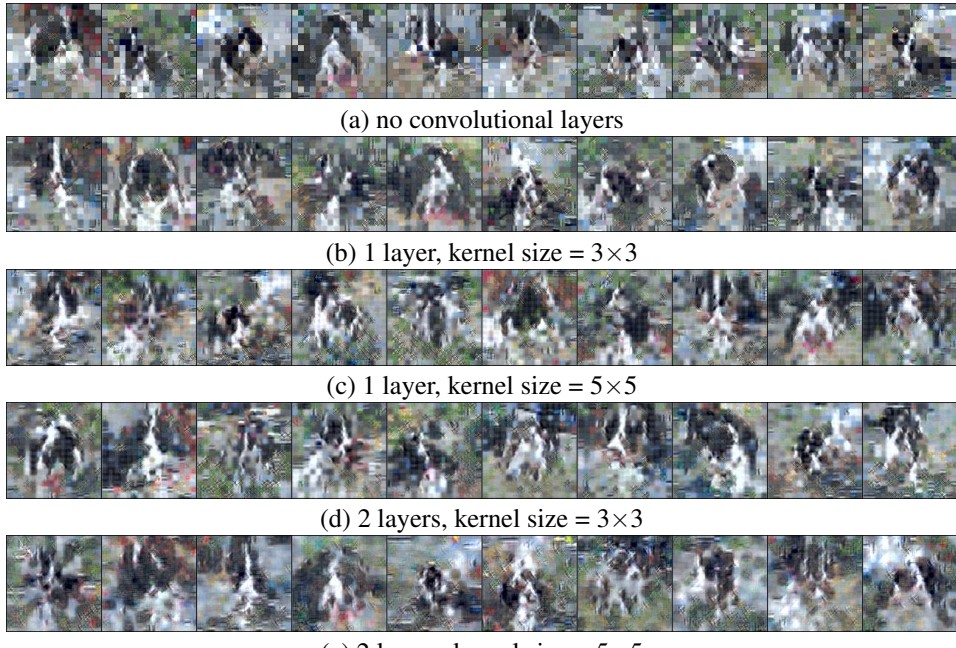

(a) no convolutional layers

(b) 1 layer, kernel size = 3×3

(c) 1 layer, kernel size = 5×5

(d) 2 layers, kernel size = 3×3

(e) 2 layers, kernel size = 5×5

Figure IV: Visualizations (border terrier class in ImageNette) regarding the incorporation of convolutional layers after the rearrangement of patches. The presence of the chessboard artifact gradually diminishes with increases in the number of convolutional layers and the size of convolutional kernels.

budget is stringent, *i.e.*, for small $k$. This indicates that our global feature sparsification technique produces synthetic images with more classifiable information under the same storage budget, making it more efficient for dataset distillation. Similar to the conclusion drawn in Sec. 3.3, we observe that as $k$ increases to a certain extent, the accuracy reaches an upper bound and then fluctuates.

**Test Accuracy vs. Distilling Time.**  We observe the test accuracy and distilling time on CIFAR10 with IPC 1 storage space, and compare with the previous methods [16, 21, 22], as illustrated in Fig. II (right). Overall, SPEED consistently outperforms previous methods when compared at similar distilling time points.

**Visualization of SCMs.**  We visualize our SCMs with varying values of $k$ in Fig. III, where the patch corresponds to zero column vector in its SCMs is marked grey. As $k$ decreases, we observe a higher preference for column vectors that correspond to background patches to be zero vectors in SCMs. This indicates that SPEED automatically learns to allocate more storage budget to foreground objects that contain more useful clues for classification. Even when $k$ is set as low as 12, the salient foreground object is still well synthesized. It should be noted that the zeroed-out patches do not occupy any storage budget since the SCMs are saved in the COO compressed format.

**Incorporation of Additional Convolutional Layers.**  We find a chessboard (blocky) artifact on synthetic images of high-resolution datasets. To investigate its influence on test accuracy, we perform multiple experiments on ImageNette with IPC 1 storage space, adding 1 and 2 convolutional layers with kernel sizes 3 and 5, as shown in Tab. II. As evident from the results, the incorporation of additional convolutional layers in our experiments does not yield a significant

| Kernel size | 3×3 | 5×5 |
|---|---|---|
| 1 layer | 66.3±1.8 | 66.4±1.3 |
| 2 layers | 65.9±1.3 | 64.0±0.5 |
| None | **66.9±0.7** | |

Table II: Enlargement of the image resolution on ImageNette.

improvement in downstream training. However, it does provide slight relief from the chessboard artifact, as depicted in Fig. IV. The impact of the chessboard artifact on downstream training and the exploration of parameter-efficient methods to eliminate these artifacts warrant further investigation.

| Spar. | Block-shared | | Evaluation | | | | |
|---|---|---|---|---|---|---|---|
| | SAET | SCM | # Param | ConvNet | MLP | ResNet18 | ViT |
| - | - | - | 27M | 41.2±0.4 | 15.6±0.2 | **29.8±0.5** | **21.1±0.3** |
| - | ✓ | - | 27M | **41.3±0.2** | **15.6±0.3** | 29.4±0.4 | 20.6±0.1 |
| - | ✓ | ✓ | 14M | 41.2±0.4 | 15.3±0.1 | 29.6±0.6 | 20.9±0.4 |
| ✓ | - | - | 549K | 40.0±0.6 | 15.6±0.1 | **30.3±0.5** | 20.7±0.4 |
| ✓ | ✓ | - | 543K | **40.3±0.7** | **15.6±0.1** | 29.7±0.7 | **21.0±0.4** |
| ✓ | ✓ | ✓ | **305K** | 40.0±0.4 | 15.5±0.2 | 29.9±0.3 | 20.7±0.5 |

Table III: Storage efficiency study. The checkmark ✓ in the Spar. column indicates the use of sparsified SCMs, while the absence of a mark indicates the use of full-element SCMs. The checkmark ✓ in the Block-shared SAET/SCM column indicates sharing SAETs/SCMs across different scales of blocks, while the absence of a mark indicates using block-specific SAETs/SCMs.

| IPC | 1 | 10 | 50 |
|---|---|---|---|
| Parameters / Class | 3,072 | 30,720 | 153,600 |
| IDC [19] | - | 40 (44.8±0.2) | - |
| HaBa [21] | 5 (33.4±0.4) | 45 (40.2±0.2) | 245 (47.0±0.2) |
| RTP [22] | 16 (34.0±0.4) | 232 (42.9±0.7) | - |
| SPEED | 11 (**40.0±0.4**) | 62 (**45.9±0.3**) | 100 (**49.1±0.2**) |

Table V: Comparisons on quantity ($N/c$) and quality (test accuracy) of synthetic images.

**Storage Efficiency Study.** To investigate the storage efficiency of recurrent blocks that involves utilizing block-shared SAETs and SCMs, we attempt to allocate specific SAETs and SCMs for each block, along with setting a uniform value for $k$ in feature sparsification, as studied in Tab. III. Our results indicate that applying recurrent blocks leads to the highest efficiency. While the block-specific strategy shows a marginal improvement over the block-shared strategy with or without feature sparsification, it lacks a significant performance advantage. However, recurrent blocks using block-shared SAETs and SCMs come with significantly lower storage, saving about $44\%$ parameters while achieving competitive results. Furthermore, we observe that feature sparsification can enhance both the overall cross-architecture generalization performance and storage efficiency.

**Number of Synthetic Images.** The number of our synthetic images on ImageNet subsets is summarized in Tab. IV. Our method synthesizes 15 images under the IPC 1 budget, and remarkably, it achieves performance that is competitive with other methods operating under the IPC 10 budget, *i.e.*, 10 synthetic images, while utilizing only $10\%$ of the parameters. For instance, on ImageNette, our method achieves an

| IPC | 1 | 10 |
|---|---|---|
| Parameters / Class | 49,152 | 491,520 |
| # Synthetic images | 15 | 111 |

Table IV: Number of synthetic images on ImageNet subsets.

impressive accuracy of $66.9\%$ with IPC 1, surpassing the previous state-of-the-art [37] result of $66.5\%$ achieved with IPC 10. These findings demonstrate the efficiency of our method and the high quality of the synthetic images it produces.

To further prove the above claim, we conclude the number of synthetic images on CIFAR100, compared with other parameterization work [19, 21, 22], as shown in Tab. V. As evident, while the number of our synthetic images is not the highest among all approaches, our outstanding performance clearly showcases the high quality of the synthetic images. This further emphasizes that our approach enhances performance by improving both the quality and quantity of the synthetic images. Moreover, it demonstrates the highly efficient reduction of spatial redundancy achieved by our method.

## III    Additional Visualizations

We include additional visualizations here. Fig. V-VII show the synthetic datasets for CIFAR10 and CIFAR100. Higher-resolution datasets are visualized in Fig. VIII-XV. We find that visualizations of lower-resolution datasets are more human-friendly since they contain less spatial redundancy. For high-resolution datasets, storage budgets are primarily allocated to salient foreground features, resulting in a significant reduction of useless patches. Nevertheless, the main texture and rough shape can still be recognized.

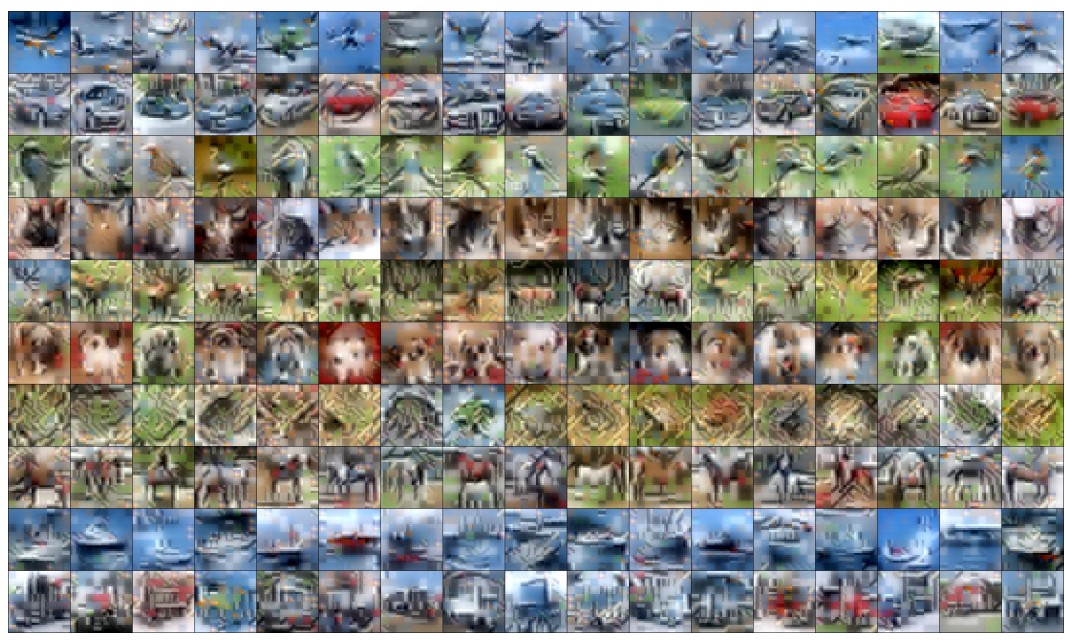

Figure V: Synthetic images on CIFAR10.

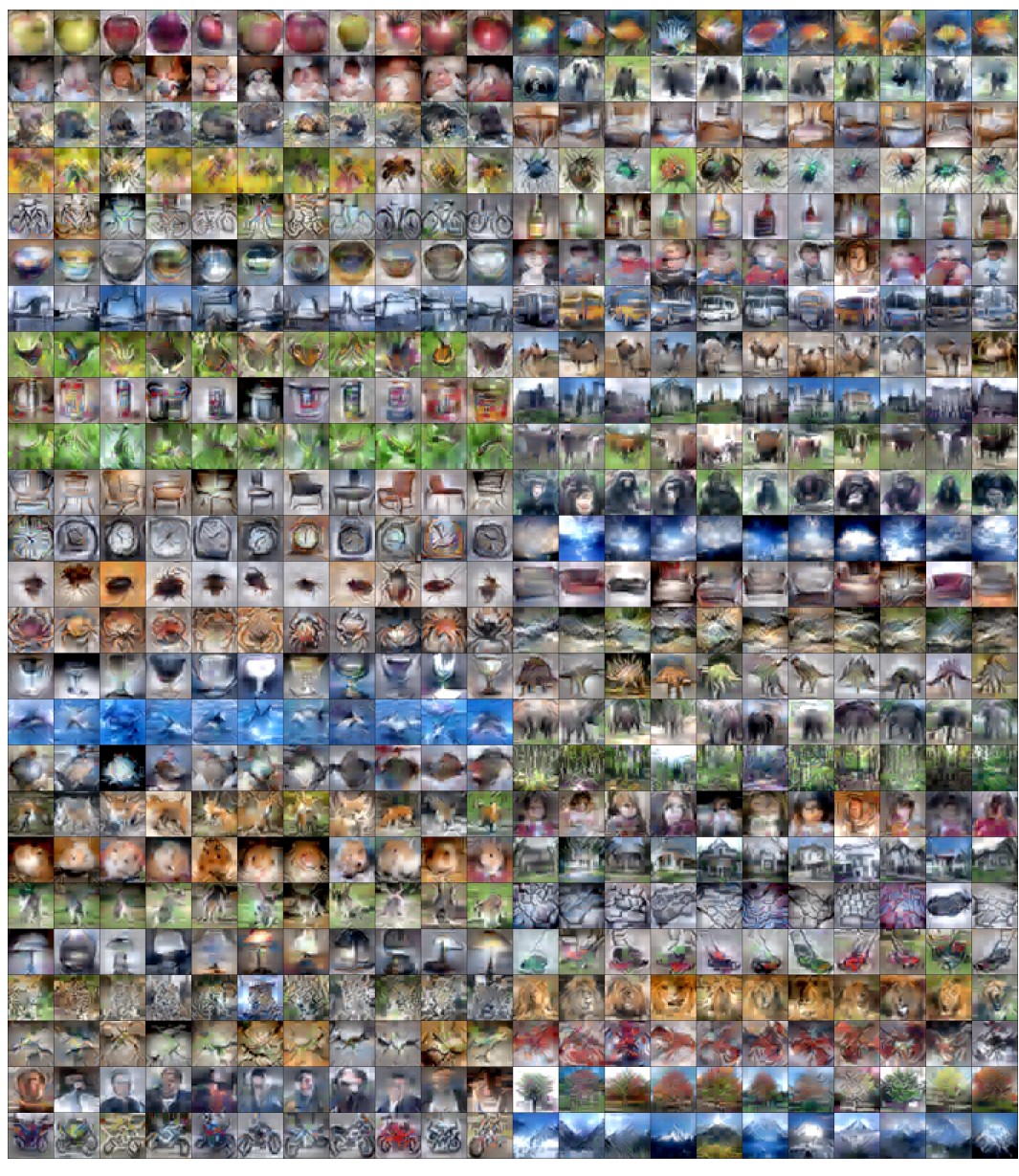

Figure VI: Synthetic images on CIFAR100. (Classes 1-50)

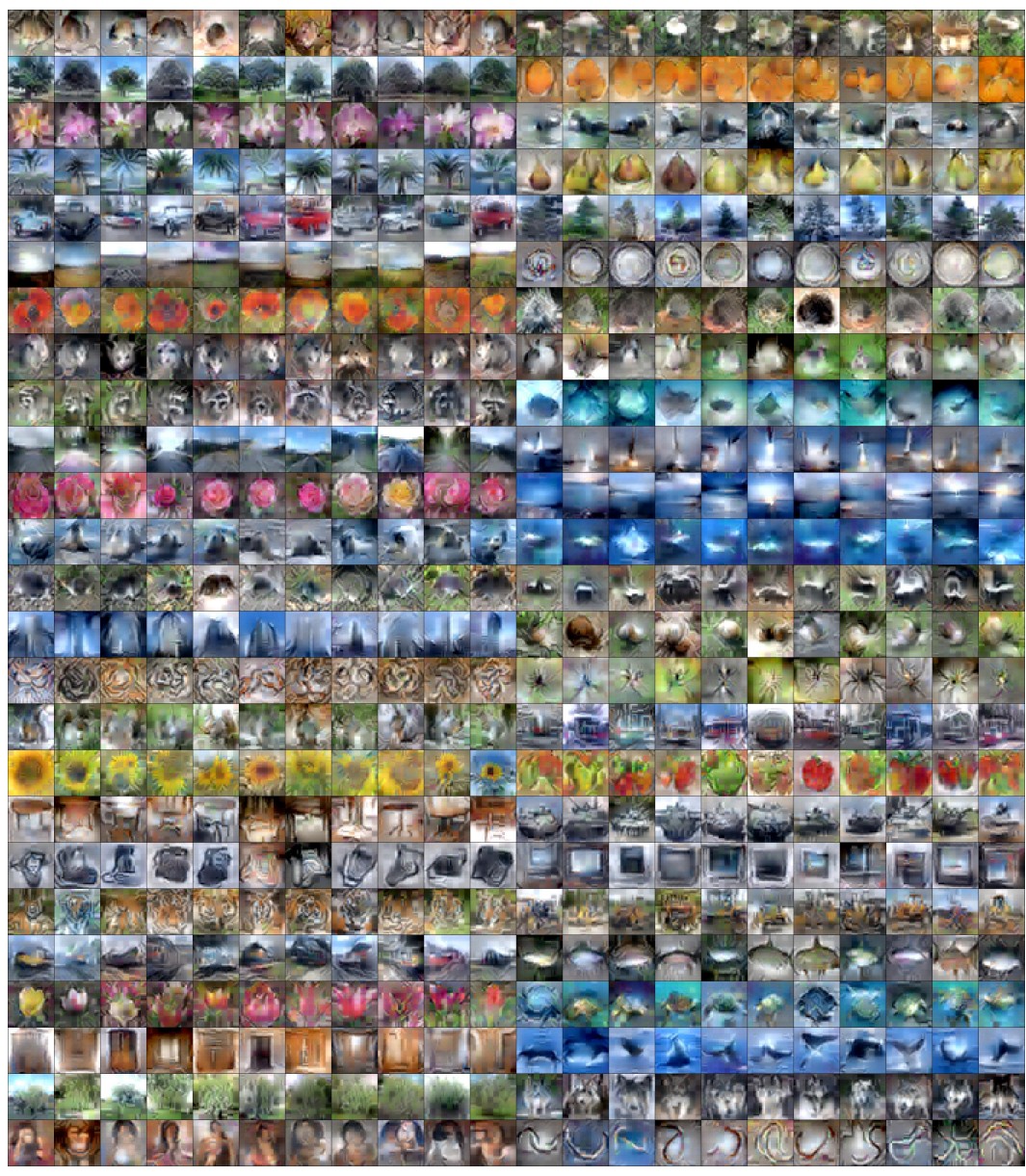

Figure VII: Synthetic images on CIFAR100. (Classes 51-100)

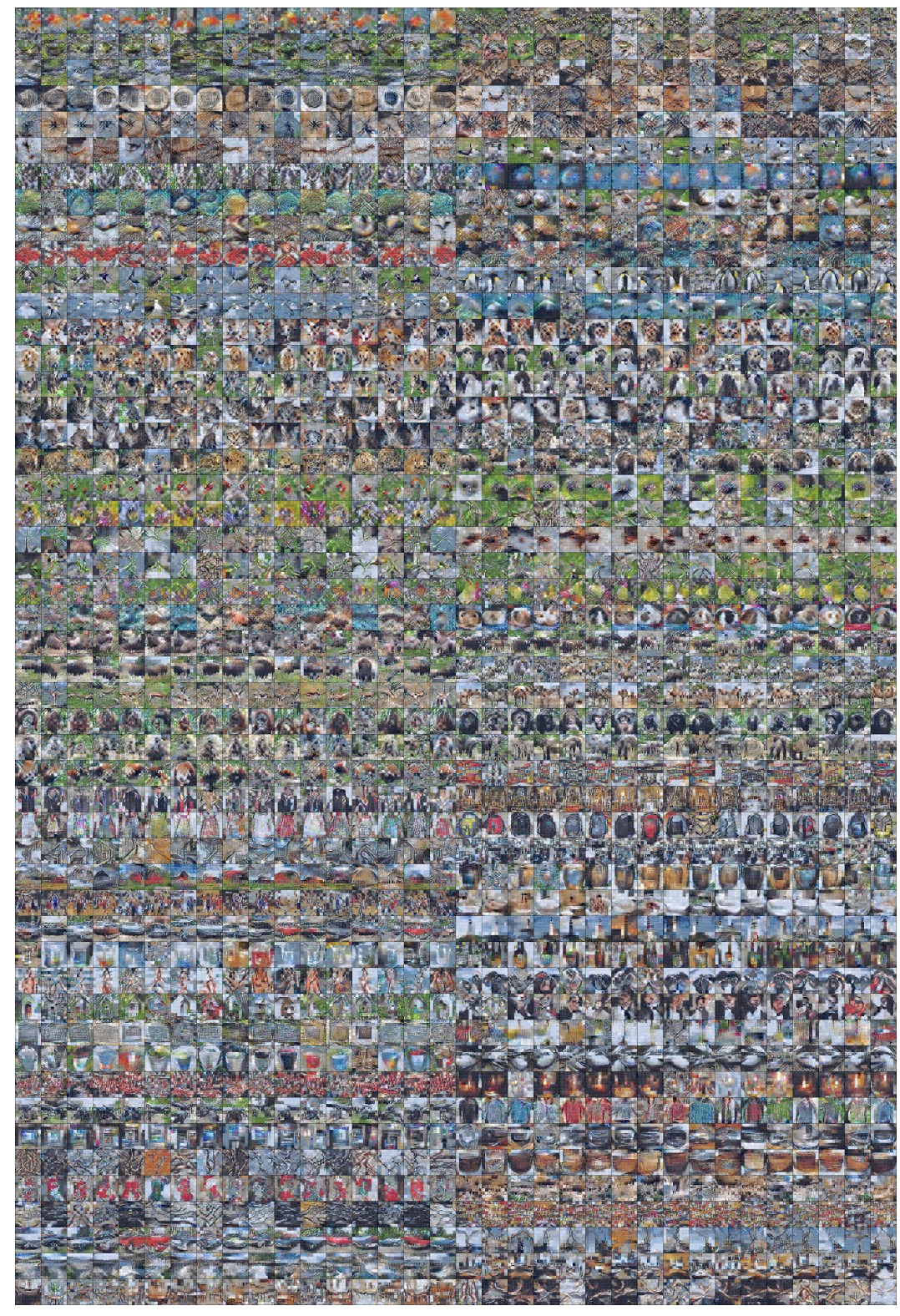

Figure VIII: Synthetic images on TinyImageNet. (Classes 1-100)

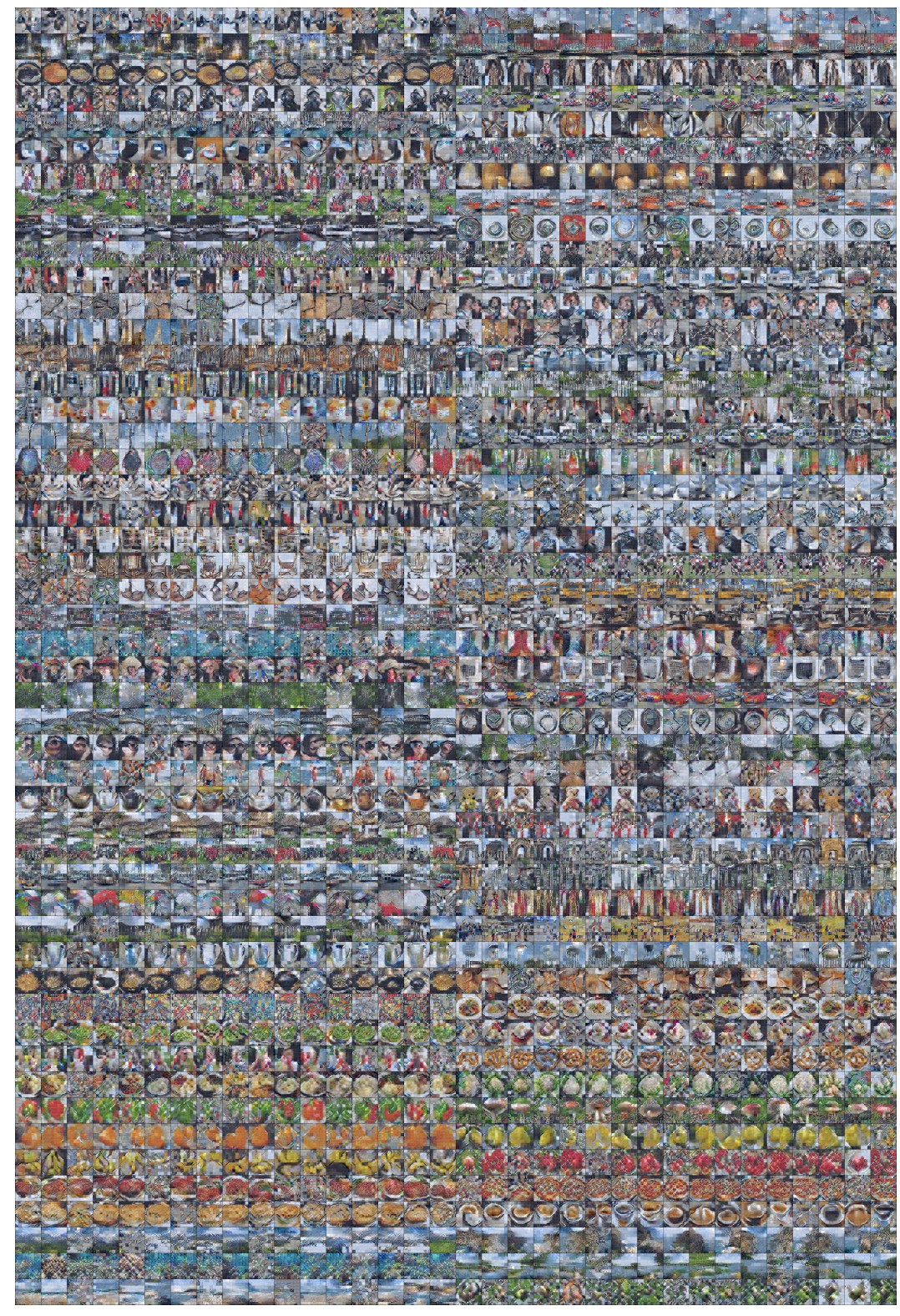

Figure IX: Synthetic images on TinyImageNet. (Classes 101-200)

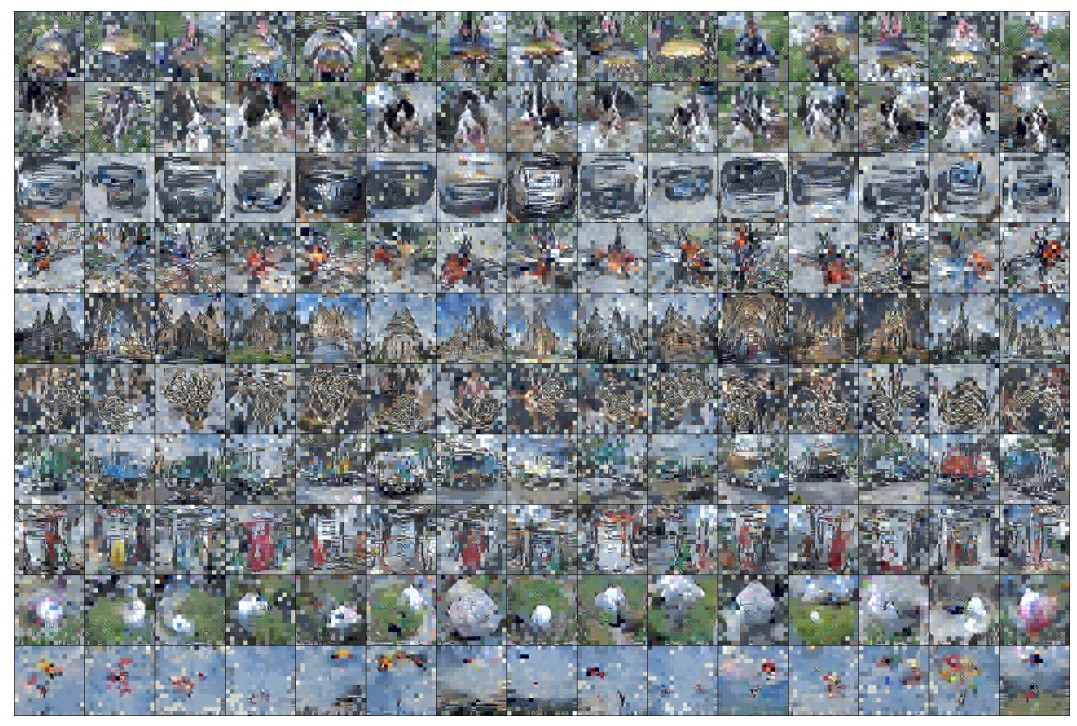

Figure X: Synthetic images on ImageNette.

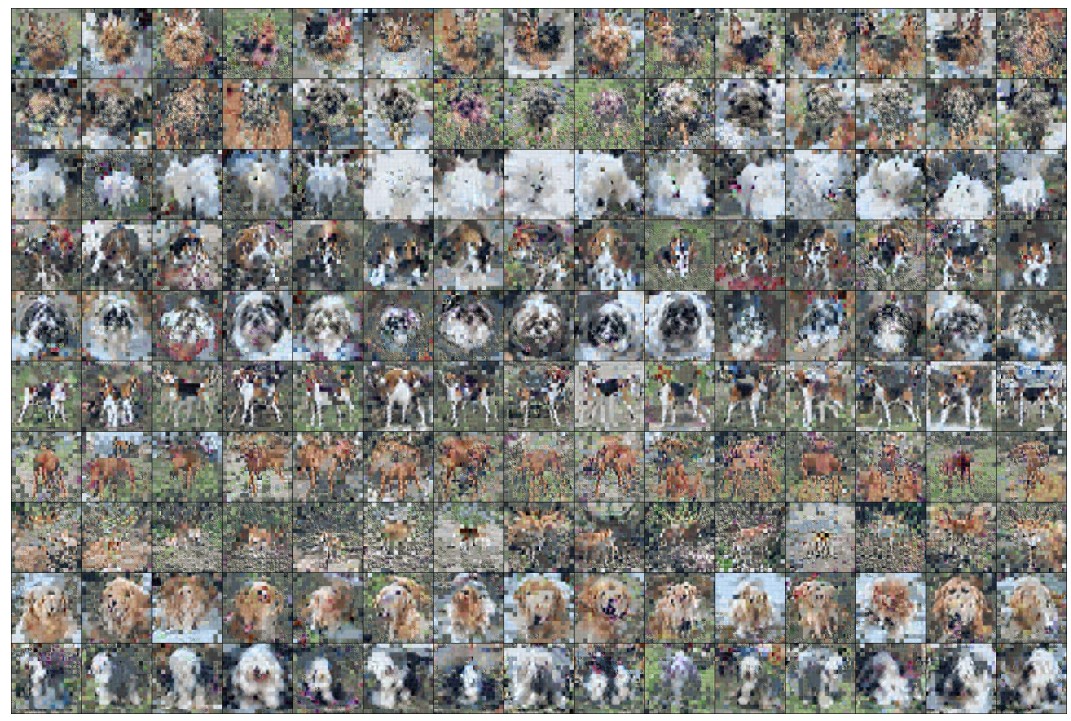

Figure XI: Synthetic images on ImageWoof.

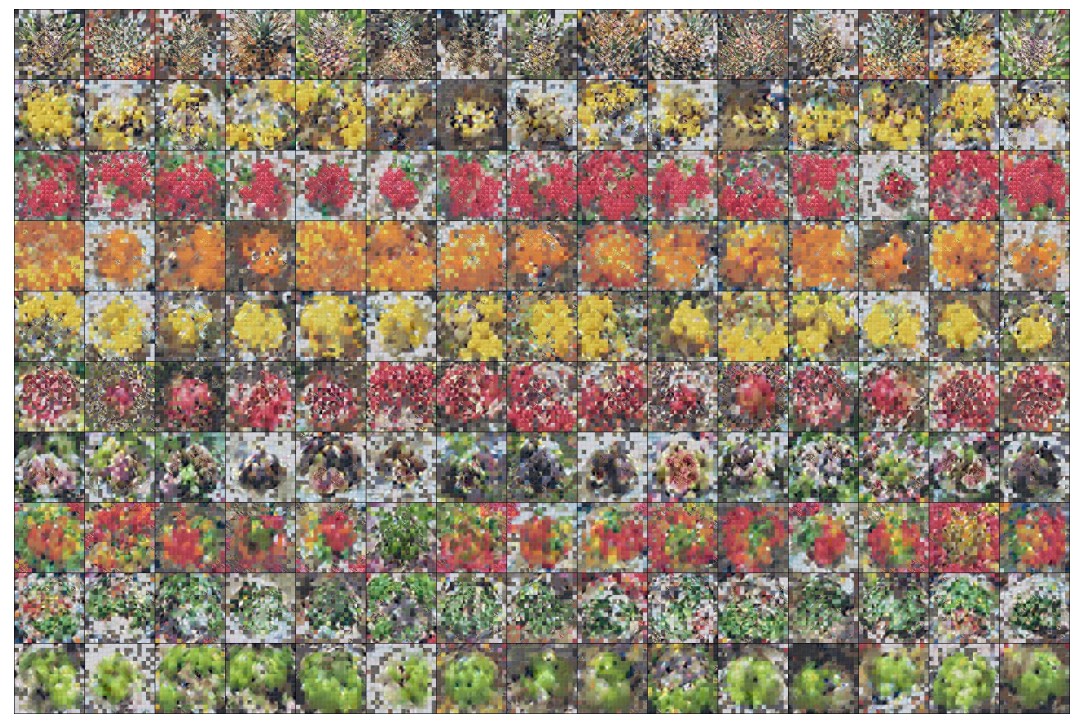

Figure XII: Synthetic images on ImageFruit.

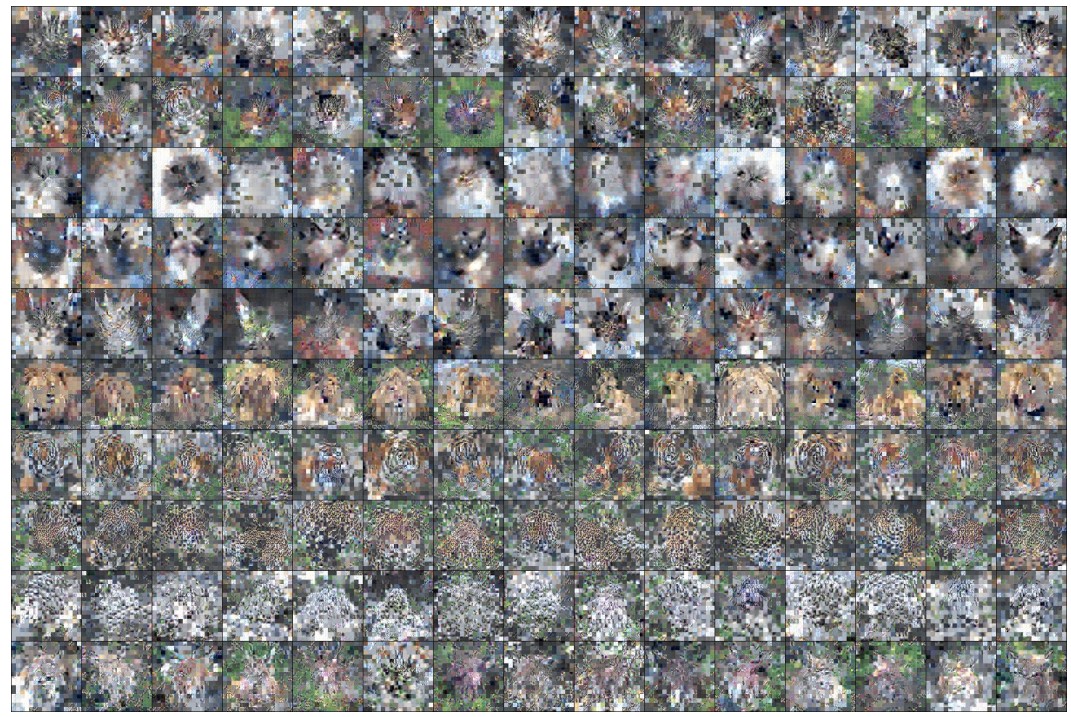

Figure XIII: Synthetic images on ImageMeow.

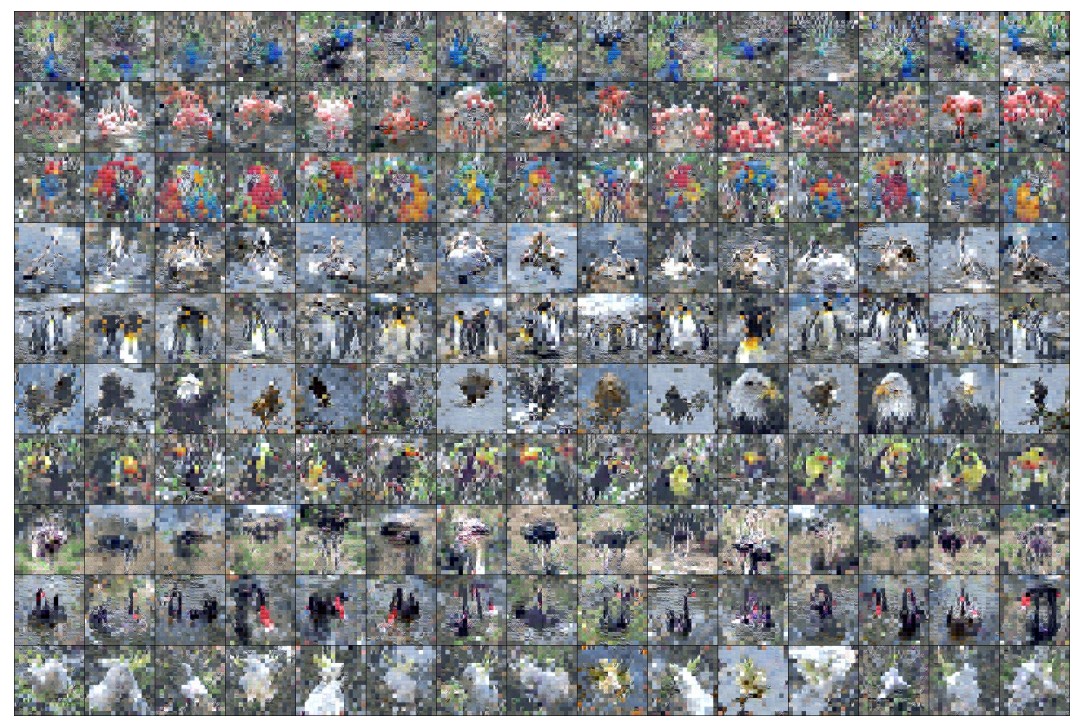

Figure XIV: Synthetic images on ImageSquawk.

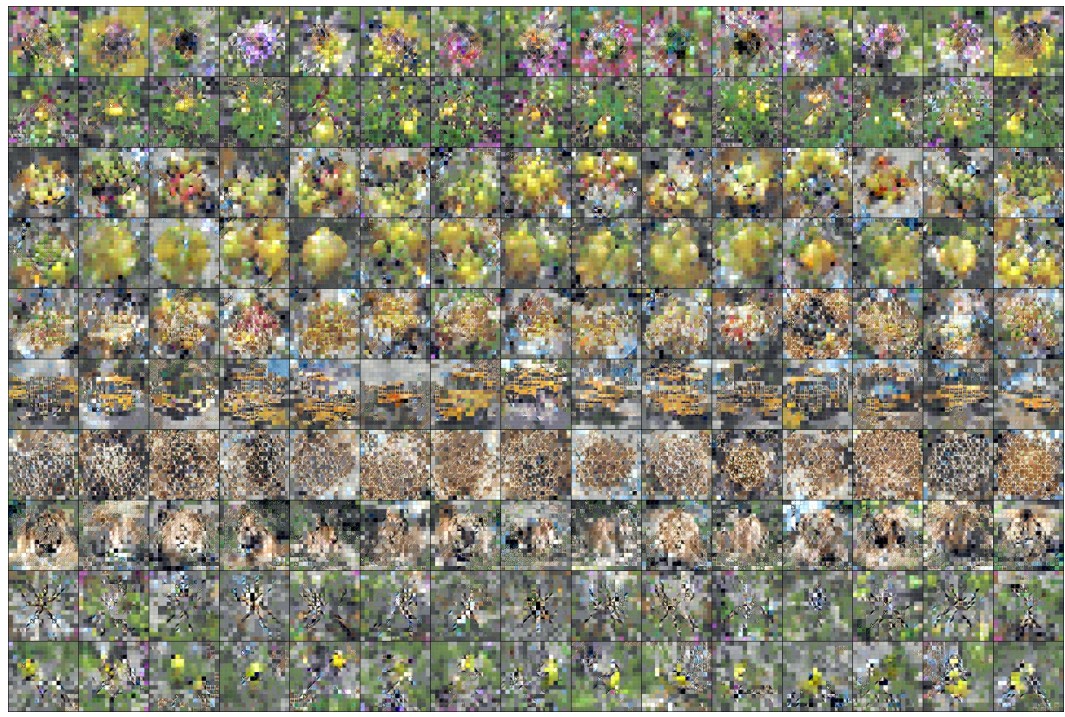

Figure XV: Synthetic images on ImageYellow.

