# OpenReview forum: "Sparse Parameterization for Epitomic Dataset Distillation"
_NeurIPS.cc/2023/Conference — NeurIPS 2023 poster_

### Official Review · Reviewer_27jp · 2023-06-22

**Soundness:** 3 good
**Presentation:** 4 excellent
**Contribution:** 3 good
**Rating:** 8
**Confidence:** 4

**Summary:**

This paper introduces the insight of sparse coding into dataset distillation and proposes a sound method. This work can efficiently generate syn data by adopting the multi-head SCMs as the shared source of the syn images and using a recurrent model to generate the syn patches. It can cooperate with various previous matching methods. In extensive experiments, the proposed method shows auspicious results and superiority.

**Strengths:**

+ Importing the sparse coding into the syn data is interesting and effective from the experiments. The analysis of the problem of overlooking the syn data sparsity itself is insightful and naturally leads to the proposed method.

+ Good writing, easy to follow.

+ Extensive experiments, ablations, and visualizations.

+ Insightful and valuable discussions and analyses, especially the syn data property, clear formulation, storage analysis, and trade-off between quality and quantity.

+ Fig III in the appendix clearly showcases the effects of the proposed method.

**Weaknesses:**

- More discussions about the relations between dataset distillation, sparse coding, and corset selection.

- Though the recurrent model is sound according to the cost discussion. I still wonder about the performance of adopting a heavier model following the same idea of the proposed method.

- Fig. 4: should be more self-contained, please explain the comparison and differences/similarities.

**Questions:**

1. If the image size is larger, what will happen?

2. Some other possible choices of the SCM format and utilization? For future study.

**Limitations:**

Discussed.

---

> ### Author Rebuttal · Authors · 2023-08-09
>
> We appreciate reviewer 27jp for the insightful and constructive comments and are glad that the reviewer finds our method novel and interesting. In response to the concerns raised, we will address them as follows:
>
> 1. **More discussions about the relations between dataset distillation, sparse coding, and coreset selection.**
>
>    * Thanks for the suggestion. As mentioned in _Lines 303 to 305_ of our paper, previous research in sparse coding has primarily focused on compressing individual images [71, 72], with the primary objective of achieving high compression ratios [34] while minimizing perceptual distortion. In contrast, dataset distillation involves condensing informative information from the original dataset into a smaller synthetic dataset to enhance downstream training without explicitly considering perceptual distortion. However, it is worth noting that many techniques and theories in sparse coding can provide valuable inspiration for developing parameterization methods in dataset distillation.
>
>    * Coreset selection [a, b, c, d] aims to identify a representative subset of the original dataset that aligns with the objectives of dataset distillation. As shown in [e], coreset selection typically performs better when the storage budget is relatively large, while dataset distillation demonstrates superior performance under extremely limited storage budgets.
>
>    * We are glad to incorporate more discussions on sparse coding and coreset selection in the paper. Additionally, we have included the results of the coreset methods in the experimental comparison, as presented in _Table A in the submitted PDF of the global response_.
>
> 2. **Though the recurrent model is sound according to the cost discussion. I still wonder about the performance of adopting a heavier model following the same idea of the proposed method.**
>
>    * Thanks for your interest in our model design. In _Table II in the supplementary material_, We compare the recurrent model (the last row of the table) and the heavier non-recurrent model (the penultimate row of the table). Further discussion on this comparison can be found in _Lines 125 to 127 of the supplementary material_. In short, a heavier model leads to a notable escalation in the number of parameters (from 305k to 543k) while yielding only a marginal improvement in accuracy (from 40.0% to 40.3%). This experiment provides evidence supporting the efficiency of our recurrent design.
>
> 3. **Fig. 4: should be more self-contained, please explain the comparison and differences/similarities.**
>
>    * Thanks for the suggestion. As mentioned in _Lines 248 to 253_, the purpose of Figure 4 is to demonstrate that the process of sparsification does not lead to a significant loss of essential information. Consequently, the two images (before and after sparsification) appear highly similar. To enhance clarity, we will update the caption of Figure 4 to include this conclusion.
>
> 4. **If the image size is larger, what will happen?**
>
>    * Thanks for the question. As stated in _Line 68_, our method is designed to efficiently handle high-resolution datasets. However, it is worth noting that previous methods did not specifically perform experiments on larger image sizes, with a maximum size of 128x128. Therefore, conducting a direct comparison with these methods on image sizes of 256x256 is not feasible.
>
>    * Following your suggestion, we performed experimental investigations on both the distribution matching baseline [17] and our method, using the ImageNette dataset with image sizes of 128x128 and 256x256. In line with the practices of previous methods that handle higher resolution images with deeper networks, we increased the depth of the ConvNet to 6 for the 256x256 image size:
>
>      ||128 x 128|256 x 256|Gain|
>      |-|-|-|-|
>      |Baseline|28.6 $\pm$ 0.6%|29.5 $\pm$ 1.1%|+0.9%|
>      |Ours|53.5 $\pm$ 1.2%|57.7 $\pm$ 0.9%|+4.2%|
>      |Gain|+24.9%|+28.2%|+3.3%|
>
>         The results demonstrate that our method achieves more substantial improvements when applied to higher-resolution images. These findings will be thoroughly incorporated into the paper, as they contribute to the expanding body of evidence supporting the effectiveness of our method on high-resolution image datasets.
>
> 5. **Some other possible choices of the SCM format and utilization? For future study.**
>
>    * Thanks for the suggestion. Several other efficient sparse matrix storage formats, such as compressed sparse row (CSR), compressed sparse column (CSC), diagonal (DIA), and block compressed sparse row (BSR), have the potential to save more storage compared to the naive coordinate (COO) format. However, DIA and BSR only achieve significant storage savings when the sparse matrix exhibits certain structural patterns. In the context of our work, the sparsity induced by the $l_1$ penalty is unstructured. As a result, we have opted for the simplest COO format to accommodate this unstructured sparsity. For CSR and CSC formats, we can convert directly from COO to them, which can bring further storage savings when the sparsity is not extremely high. We will consider other sparsity penalties and their corresponding efficient storage formats in our future work.
>
> ----
>
> [a] Welling, Max. "Herding dynamical weights to learn." Proceedings of the 26th Annual International Conference on Machine Learning. 2009.
>
> [b] Chen, Yutian, Max Welling, and Alex Smola. "Super-samples from kernel herding." arXiv preprint arXiv:1203.3472 (2012).
>
> [c] Feldman, Dan, Matthew Faulkner, and Andreas Krause. "Scalable training of mixture models via coresets." Advances in neural information processing systems 24 (2011).
>
> [d] Sylvestre-Alvise Rebuffi et al. “icarl: Incremental classifier and representation learning”. Proceedings of the IEEE conference on Computer Vision and Pattern Recognition. 2017.
>
> [e] Cui, Justin, et al. "DC-BENCH: Dataset condensation benchmark." Advances in Neural Information Processing Systems, 2022.

---

> > ### Comment · Reviewer_27jp · 2023-08-14
> > **Post-rebuttal**
> >
> > Thank the authors for the response.
> >
> > My main concerns are addressed well in the above rebuttal. However, the other reviewers have shared some important concerns which should be discussed, also looking forward to their opinions. Overall, I think this paper proposed an interesting combination of two directions and has good inspirations.

---

### Official Review · Reviewer_7J1t · 2023-07-05

**Soundness:** 2 fair
**Presentation:** 2 fair
**Contribution:** 2 fair
**Rating:** 6
**Confidence:** 4

**Summary:**

The paper proposes a new framework(SPEED) to perform dataset distillation.

The new framework is composed of 3 parts:
1. Spatial-Agnostic Epitomic Tokens (SAETs)
2. Sparse Coding Matrices (SCMs)
3.  A Feature-Recurrent Network (FReeNet)

The paper also employees multi-head attention to ensure the diversity of distilled images, thus achieving new SOTA.
The new framework SPEED is also claimed to work with multiple dataset matching methods and enhance their performances.

Through various experiments, the method shows strong performances on CIFAR-10/100 and TinyImageNet. Similar results are also observed on ImageNet subsets.

### I have read the author's response and my concerns are addressed by seeing more experiment results.

**Strengths:**

## originality
SPEED is claimed to be the first paper studying the spatial redundancy in the field of dataset distillation/condensation

SPEED applies a few methods such as the concepts from ViT, dictionary learning and sparse coding.

## quality
The method works very well on dataset with higher resolution.

## clarity
The paper is well written and easy to follow

The framework of the method and learned images are visualized which makes it easy to understand.

## significance
The proposed methods achieve SOTA performances under most settings.

The paper novelly proposes to distill images at image patch level which shows a new way for dataset distillation.

The proposed method is compatible with multiple matching objectives.

**Weaknesses:**

- Incomplete evaluation results in table 2. IPC 1/10/50 are used for CIFAR-10/100, but only IPC 1 for TinyImageNet is used. Why that is the case? From [1], even the trajectory matching method this paper adopts has reported the results on TinyImageNet with IPC 1/10/50.

- In table 1, for the parameterization methods including SPEED, can the author include how many synthetic images are generated for evaluations, e.g. 11 for IPC 1 so that we know if the performance gain is due to increased number of images or increased quality of generated images.







[1] Dataset Distillation by Matching Training Trajectories

**Questions:**

See my comments in weakness section. On general, this paper provides some insights on a totally different way of parameterization in dataset distillation. I am willing to raise the score if the above questions are answered.

**Limitations:**

The paper proposes a new method to apply parameterization in dataset distillation. However, the generalization ability of the method is unknown such as the recurrent neural network. Especially some evaluation results in table 2 are missing which makes it even harder to tell.

---

> ### Author Rebuttal · Authors · 2023-08-09
>
> We sincerely thank reviewer 7J1t for the pertinent and valuable feedback. We are delighted to learn that the reviewer finds our method achieves good performance across multiple datasets demonstrated in Table 1 and Table 2. The concerns are fully addressed as follows.
>
> 1. **Incomplete evaluation results in table 2? Only IPC 1 for TinyImageNet is used. Why that is the case?**
>
>    * Thanks for the question. It is important to highlight that our results, achieved under the IPC 1 storage budget (ACC: 26.9%), have already surpassed the performance of the baseline method (trajectory matching [16]) under the IPC 10 storage budget (ACC: 23.2%) on TinyImageNet. Moreover, our results even approached the performance of the baseline method under the IPC 50 (ACC: 28.0%) setting.
>
>    * As highlighted by reviewer yNYq, it is of utmost importance to conduct comprehensive comparisons among different parameterization methods. Previous parameterization studies either lacked reporting results on TinyImageNet [19, 21] or only provided results under the IPC 1 budget [22]. Consequently, experiments conducted under higher IPCs lack meaningful comparisons. Due to the limitations of time and computational resources for submissions, we did not prioritize these experiments accordingly.
>
>    * As per your suggestion, we are willing to report our experimental results on TinyImageNet under IPC 10/50 as an addition, as shown in the following table:
>
>      | IPC (#Param) | 1 (12,288)             | 10 (122,880)             | 50 (614,400)             |
>      | -------- | ----- | ----- | -----         |
>      | TM | 8.8 $\pm$ 0.3%  | 23.2 $\pm$ 0.2% | 28.0 $\pm$ 0.3%        |
>      | IDC | -  | - | -        |
>      | HaBa | -  | - | -        |
>      | RTP | 16.0 $\pm$ 0.7%  | - | -        |
>      | Ours     | **26.9 $\pm$ 0.3%** | **28.8 $\pm$ 0.2%** |    **30.1 $\pm$ 0.3%**   |
>
>      Our method achieves an accuracy of 28.8% under the IPC 10 budget, showcasing substantial improvements over the baseline (ACC: 23.2%), and even surpassing the baseline under IPC 50 (ACC: 28.0%). Additionally, our method achieves an accuracy of 30.1% under the IPC 50 budget, establishing a state-of-the-art performance. These experimental results have been incorporated into _Table A in the PDF of the global response_, thus making our evaluation more comprehensive.
>
> 2. **Can the author include how many synthetic images are generated for evaluations? Is the performance gain due to increased number of images or increased quality of generated images?**
>
>    * Thanks for your insightful attention to the quality and quantity aspects of dataset parameterization. We also recognize the significance of this topic, and therefore, we have dedicated a single section in our paper to thoroughly discuss it. The experimental results can be found in _Tables 6 to 8_, and the discussion can be found in _Lines 254 to 270_. In conclusion, both quality and quantity play crucial roles in parameterization. Excessively sacrificing one in favor of the other without careful consideration can lead to a noticeable decline in performance.
>
>    * As suggested, the number of synthetic images on ImageNet Subsets in Table 1 is summarized:
>
>      | IPC (#Param)  | 1 (49,152) | 10 (491,520) |
>      | ---- | ----      |----|
>      | #Synthetic Images | 15 |  111  |
>
>      Our method synthesizes 15 images under the IPC 1 budget, and remarkably, it achieves performance that is competitive with other methods operating under the IPC 10 budget, while utilizing only 1/10 of the parameters. For instance, on ImageNette, our method achieves an impressive accuracy of 66.9% under the IPC 1 budget, surpassing the previous state-of-the-art result of 66.5% achieved under the IPC 10 budget. These findings demonstrate the efficiency of our method and the high quality of the synthetic images it produces.
>
>    * To further prove the above claim, we conclude the number of synthetic images on CIFAR100, compared with other parameterization work:
>
>
>      | IPC (#Param) | 1 (3,072)             | 10 (30,720)             | 50 (153,600)             |
>      | ---- | -------------- | --------------- | --------------- |
>      | IDC  | -      | 40 (44.8 $\pm$ 0.2%)      | -     |
>      | HaBa | 5 (33.4 $\pm$ 0.4%)      | 45 (42.5 $\pm$ 0.2%)      | 245 (47.0 $\pm$ 0.2%)     |
>      | RTP  | 16 (34.0 $\pm$ 0.4%) | 232 (42.9 $\pm$ 0.7%)     | -     |
>      | Ours | 11 (**40.0 $\pm$ 0.4%**)     | 62 (**45.9 $\pm$ 0.3%**) | 100 (**49.1 $\pm$ 0.2%**) |
>
>     As evident, while the number of synthetic images generated by our method is **not the highest** among all approaches, our outstanding performance clearly showcases the **high quality** of the synthetic images. This further emphasizes that our approach enhances performance by improving both the quality and quantity of the synthetic images. Moreover, it demonstrates the highly efficient reduction of spatial redundancy achieved by our method.
>
> 3. **The generalization ability of the method is unknown such as the recurrent neural network.**
>
>    * We appreciate the reviewer's attention to the generalization abilities of our method. We have discussed and proved the generalization abilities of our method in _Section 3.2 (Lines 203 to 232)_, including cross-architecture performance, universality to matching objectives, and robustness to corruption. To gain a comprehensive understanding of the experimental results and delve into an in-depth analysis, we kindly refer the reviewer to _Section 3.2_ of the paper.
>
>    * For the ablation study of our recurrent blocks, please kindly refer to _Table II in the supplementary material_ and the second question of reviewer 27jp.

---

> > ### Comment · Reviewer_7J1t · 2023-08-14
> > **thanks for the response**
> >
> > thanks for the authors' response. The results look good to me. Please include these into the paper. I recommend acceptance for this paper after reviewing the results.

---

### Official Review · Reviewer_yNYq · 2023-07-06

**Soundness:** 4 excellent
**Presentation:** 3 good
**Contribution:** 4 excellent
**Rating:** 7
**Confidence:** 5

**Summary:**

This work proposes a new memory-saving method of dataset distillation by distilling the dataset into a set of Spatial-Agnostic Epitomic Tokens which are indexed by Sparse Coding Matrices and decoded into images by a Feature-Recurrent Network. This method is plug-and-play compatible with existing distillation methods, allowing them to achieve more efficient storage. State-of-the-art results are shown for many datasets and problem settings.

**Strengths:**

The overall presentation of the paper is very nice. The figures and equations very clearly explain to the reader the main ideas. The colorfully annotated Eq 7 especially makes the storage budget easy for the reader to digest.

The SPEED method itself is quite interesting, and algorithm 1 very clearly explains the process.

The many ablation studies and side-experiments further explain the effectiveness of the method.

**Weaknesses:**

The biggest issues I have with this paper are the presentation of tables 1 and 2.

This is not an issue specific to this paper, but methods that do re-paramaterization as a means of memory saving should not be directly compared to methods that only propose a matching algorithm.

It should be made extremely clear that IDC, HaBa, RTP and SPEED are solving an inherently different problem than the baseline methods.

Synthetic set size should also not be given in "IPC" but in the number of learnable parameters, since the given IPC simply isn't true anymore. For example, instead of IPC=10, you could have #Params <= 30,720 (10x3x32x32)

**Questions:**

I am happy to raise my score if the above weakness is addressed.

I am also just curious about another thing:

The high resolution images seem extremely blocky due to the nature of the patches.

Have you considered adding 1 or 2 convolutional layers _after_ the patches are stitched back together?

**Limitations:**

yes

---

> ### Author Rebuttal · Authors · 2023-08-09
>
> We appreciate reviewer yNYq for the insightful suggestions and are happy that the reviewer finds our work interesting and effective. We are glad to address the concerns and take the suggestions as follows:
>
> 1. **The presentation of tables 1 and 2. Re-parameterization should not be directly compared to methods that only propose a matching algorithm.**
>
>    * Thank you for considering the suggestion. We acknowledge that a direct comparison between pure matching methods and parameterization methods can be a matter of controversy, and we agree on the importance of presenting the tables in a manner that eliminates any ambiguity. As per your request, we have reorganized Table 2 into a _Table A in the submitted PDF of the global response_. In Table A, we have made explicit differentiations between pure matching methods and parameterization methods, ensuring a clear separation between these two categories. Furthermore, we have included an additional row to indicate the corresponding parameter amounts for the parameterization methods. We will also make the same revisions to Table 1.
>
> 2. **The high resolution images seem extremely blocky. Have you considered adding 1 or 2 convolutional layers after the patches are stitched back together?**
>
>    * We sincerely appreciate the valuable advice provided. As suggested, we performed multiple experiments on ImageNette under IPC 1 storage budget, adding 1 and 2 convolutional layers with kernel sizes 3 and 5:
>
>      | Kernel Size | 3x3       | 5x5        |
>      | ------      | ---       | ---        |
>      | 1 layer     | 66.3 $\pm$ 1.8%     |  66.4 $\pm$ 1.3%     |
>      | 2 layers     | 65.9 $\pm$ 1.3%     |  64.0 $\pm$ 0.5%     |
>      | None        | **66.9 $\pm$ 0.7%** |  **66.9 $\pm$ 0.7%** |
>
>      As evident from the results, the incorporation of additional convolutional layers in our experiments did not yield a significant improvement in downstream training. However, it did provide slight relief from the chessboard artifact (blocky artifact), as depicted in _Figure A in the submitted PDF of the global response_. Nonetheless, the impact of the chessboard artifact on downstream training and the exploration of parameter-efficient methods to eliminate these artifact warrant further investigation.

---

> > ### Comment · Reviewer_yNYq · 2023-08-14
> > **Response to Rebuttal**
> >
> > Thank you very much for addressing my concerns.
> >
> > As promised, I will raise my score to a 7.

---

### Official Review · Reviewer_kT2g · 2023-07-10

**Soundness:** 3 good
**Presentation:** 2 fair
**Contribution:** 2 fair
**Rating:** 5
**Confidence:** 4

**Summary:**

This paper proposes a new parameterization for dataset distillation. The new parameterization considers image patches, use sparse matrix and recurrent feature net to generate synthetic images. The total parameters follows storage constraint. The experimental results show improvement over previous methods.

**Strengths:**

+ This paper proposes a new parameterization for synthetic images in Dataset Distillation.
+ The proposed method works well across various benchmarks, including imagenet, cifar and cross arch generalization
+ This paper is quite interesting in proceeding the research on Dataset Distillation parameters. Spatial redundancies are quite dominant in standard parameterization, and this method can certainly help with alleviating that.

**Weaknesses:**

- The paper claims on reducing spatial redundancies. I wonder how the authors compare their method with [19]. Does that also consider reducing spatial redundancies?
- The paper provides better empirical performance, but the research messages are not quite surprising.
- Have the authors considered using algorithms similar to [a] for sparsification?

[a] Parameter-Efficient Transfer Learning with Diff Pruning

**Questions:**

See weaknesses section.

**Limitations:**

Yes

---

> ### Author Rebuttal · Authors · 2023-08-09
>
> We sincerely thank reviewer kT2g for the valuable comments and feedback. We deeply appreciate the reviewer's acknowledgment of the merits of our proposed method, including its effectiveness in reducing spatial redundancy, its capacity for generalization, and its superior performance compared to previous approaches. We are fully committed to addressing the concerns raised by the reviewer, and we outline our responses below.
>
> 1. **How the authors compare their method with [19]. Does that also consider reducing spatial redundancies?**
>
>     As we discussed in _Line 283_, IDC [19] takes a simple downsampling strategy to save more images, which also partially reduces the spatial redundancy. However, naive downsampling suffers from two major drawbacks:
>    * The uniform downsampling operator employed in IDC indiscriminately discards spatial information, including crucial details that are vital for downstream training. In contrast, our method incorporates end-to-end spatial-agnostic learning to preserve informative areas that are crucial for downstream training, while discarding irrelevant and repetitive areas.
>    * Downsampling **can not** effectively reduce spatial redundancy within the same image across different locations, let alone across different images, as it is **not spatial-agnostic**. However, our method is specifically designed to be spatial-agnostic, allowing us to further mitigate spatial redundancy in such scenarios.
>
>
> 2. **Have the authors considered using algorithms similar to [a] for sparsification?**
>
>    * We appreciate the introduction to the noteworthy work [a] on transfer learning. This method employs a more sophisticated approach to approximate $l_0$ sparsity by relaxing a binary vector into continuous space. In contrast, our method directly utilizes $l_1$ penalty as a surrogate for $l_0$ penalty. Both techniques are widely recognized and extensively utilized in the machine learning community.  It is important to note that the specific method used to approximate $l_0$ sparsity does not affect our overall parameterization framework.  We will discuss various sparsification techniques in our related work, including the one presented in [a], and plan to investigate and evaluate their respective impacts in future studies. We appreciate your suggestion.
>
> ----
>
> [a] Guo, Demi, Alexander M. Rush, and Yoon Kim. "Parameter-efficient transfer learning with diff pruning." arXiv preprint arXiv:2012.07463 (2020).

---

### Author Rebuttal · Authors · 2023-08-10

We would like to appreciate all the reviewers for their time and effort in the review process. Overall, we are pleased that the reviewers recognize the novelty (reviewer 27jp, yNYq, 7J1t), impressive experimental results (reviewer kT2g, 7J1t), and clear presentation (reviewer 27jp, yNYq, 7J1t) of this work. Please refer to the attached PDF for tables and figures.

---

### Comment · Area_Chair_BSVF · 2023-08-18
**Please look at the authors' reply**

Dear Reviewers,

Please do look at the authors' rebuttal if you have not done so. Please let the authors know if they have addressed your concerns.

Thanks for your contribution to NeurIPS.

AC

---

### Decision · Program_Chairs · 2023-09-21

**Decision:**

Accept (poster)

**Comment:**

The manuscript has been reviewed by four reviewers, all of whom recommended the acceptance of the manuscript.

Essentially, the reviewers found the proposed approach interesting, the paper well organized, and the experiments to be sufficient.

As such, there is no basis to overturn the consensus of the reviewers. The AC recommends acceptance.

Congrats!